# Galvanically Stimulated Degradation of Carbon-Fiber Reinforced Polymer Composites: A Critical Review

**DOI:** 10.3390/ma12040651

**Published:** 2019-02-21

**Authors:** Stanley Udochukwu Ofoegbu, Mário G.S. Ferreira, Mikhail L. Zheludkevich

**Affiliations:** 1Department of Materials and Ceramic Engineering, CICECO-Aveiro Institute of Materials, University of Aveiro, Campus Universitário de Santiago, 3810-193 Aveiro, Portugal; mgferreira@ua.pt; 2MagIC, Institute of Materials Research, Helmholtz-Zentrum Geesthacht, Max-Planck-Strasse 1, 21502 Geesthacht, Germany; mikhail.zheludkevich@hzg.de; 3Institute for Materials Science, Faculty of Engineering, Kiel University, 24103 Kiel, Germany

**Keywords:** carbon fiber, polymer matrix, metals, multi-material, interface, cathodic, galvanic coupling, high pH, degradation

## Abstract

Carbon is used as a reinforcing phase in carbon-fiber reinforced polymer composites employed in aeronautical and other technological applications. Under polarization in aqueous media, which can occur on galvanic coupling of carbon-fiber reinforced polymers (CFRP) with metals in multi-material structures, degradation of the composite occurs. These degradative processes are intimately linked with the electrically conductive nature and surface chemistry of carbon. This review highlights the potential corrosion challenges in multi-material combinations containing carbon-fiber reinforced polymers, the surface chemistry of carbon, its plausible effects on the electrochemical activity of carbon, and consequently the degradation processes on carbon-fiber reinforced polymers. The implications of the emerging use of conductive nano-fillers (carbon nanotubes and carbon nanofibers) in the modification of CFRPs on galvanically stimulated degradation of CFRP is accentuated. The problem of galvanic coupling of CFRP with selected metals is set into perspective, and insights on potential methods for mitigation and monitoring the degradative processes in these composites are highlighted.

## 1. Introduction

Besides their predominant usage in the transport industry, carbon-fiber reinforced polymers (CFRPs) are employed in other technological applications in a wide variety of industries [1]. In the medical industry, they used in such applications as fracture fixation devices and orthopedic implants [2,3,4,5,6,7,8,9,10,11,12,13,14,15,16]. In the construction and civil engineering industry, CFRP is used as a structural member in construction and for retrofitting and repairing structural deficiencies in infrastructures such as bridges and buildings caused by age, corrosion, mis-use, and/or seismic disturbances [17,18,19,20,21,22,23,24,25,26,27,28,29,30,31,32,33,34,35]. CFRP is also used as an electromagnetic interference shield [36,37,38], electric discharge, and lightening strike protection on modifications to enhance electrical conductivity [39], sports products [40], in the maritime industry in boats, ships, and submarines [41,42,43,44], and as a corrosion resistant and reinforcing liner in pipes. In the offshore oil and gas industry it is used in the repair of corroded and mechanically-damaged onshore pipelines in which the predominant load is internal pressure [45,46,47,48,49,50,51]. In addition to good strength-to-weight ratios of CFRP’s, a potential for self-healing (i.e., recovery of mechanical and other properties post damage) has been demonstrated [52,53].

Besides its use in the applications above, CFRP has the potential to become a multi-functional material capable of being deployed for simultaneous structural and sensing applications. This outlook is based on the multiple reports of the sensing ability of CFRP electrodes for a wide variety of compounds. Carbon–polyvinylchloride (C–PVC) composite electrode has been reported [54] to detect dopamine, ascorbic acid, uric acid, and their mixtures. Thostenson and Chou [55,56] have demonstrated real-time in-situ sensing of the onset, nature, and evolution of damage in advanced fiber reinforced composites with an epoxy matrix.

An interesting development and emerging application that could consolidate, and greatly increase the consumption and wider usage of CFRP is its processing by additive manufacturing. Additive manufacturing can enable easier processing of stronger products with better weight optimization and highly orientated properties guided by information on anticipated load conditions [57,58,59,60,61,62,63,64,65]. Recent reports [60,61,62,63,64,65] of successful 3D printing and additive manufacturing of CFRP parts are capable of tilting the economics of weight-to-strength optimization strategies using CFRP, encouraging its use in more applications due to reduced costs and greater ease in its processing that can allow low production volumes of parts with more intricate geometrical complexities that precludes their production with current CFRP manufacturing methods. On the strength of our evaluation of the current use and applications of CFRP, the regulatory environment (both external and internal) for reduction of CO_2_ emission in the transport industry, and the current and emerging technologies for CFRP processing, a rapid increase in CFRP consumption and its adoption in a wider range of applications is projected. This will be due to a favorable regulatory environment and emerging manufacturing processes (3D printing and/or additive manufacturing) that is very likely to significantly lower production costs while enabling the production of more intricate parts at lower production volumes. 

In most of its technological applications CFRPs are used as component(s) of hybrid structures. Multi-material combinations containing CFRPs, which are common in weight-optimized structures used in the aeronautical and other transport industries, are seeing progressive increases in the percentage of CFRP employed in these multi-material structures and in the number of the other constituent materials [66,67,68,69,70,71,72,73,74,75,76,77,78,79,80,81,82]. Hybrid structures consisting of CFRPs can emerge in the joining of CFRP with other materials such as in the adhesive attachment of fiber-reinforced polymers (FRP) laminates to the external face of reinforced concrete structures while retrofitting and strengthening concrete structures [83], in joining of CFRP laminates to metals where galvanic coupling can occur due to failure of insulating liners and/or poor joining or due to age [84,85,86,87,88,89,90,91]. More examples of such hybrid structures containing CFRP abound in the design of lightening protection systems for aircraft structure in which a continuous layer of conductive perforated metal foil or a metal honey-comb structure is consolidated in a polymer composite [92,93,94,95], or polymer composite surface is in contact with very conductive coatings most probably composed of metal particles [96]. Irrespective of the application or the process by which galvanic coupling of CFRP with metals is established, the carbon fibers in CFRPs being conductive are able to support electrochemical charge transfer processes leading to anodic dissolution of the metal, and possible sabotage of structural integrity. This phenomena can be quite deleterious if the ratio of the cathodic surface area (CFRP) is much higher than the metallic component (the anode). Consequently an appreciation of the processes involved when CFRPs are galvanically coupled to metals is vital to developing strategies for multi-material corrosion mitigation in hybrid structures containing CFRPs. These processes occur on CFRP surface and can have degradative effects on the CFRP itself, and metals galvanically coupled to it. These cathodic processes, their degradative effects and possibilities for monitoring them shall be reviewed. In addition current research efforts at modifying CFRPs to enhance mechanical properties and its implications for multi-material corrosion are highlighted.

## 2. Composition of Carbon-Fiber Reinforced Polymers (CFRPs)

Carbon-fiber reinforced polymers (CFRPs) is a name used to refer to a composite material composed of carbon fibers as the reinforcing face and a matrix composed of a polymer which can be a themosetting or a thermoplastic polymer. Besides the principal constituents; reinforcing carbon fibers and the polymer resins, other constituents can be added to enhance mechanical properties and processability. Coupling agents and coatings/treatments [97] on the carbon fibers and fillers in the matrix can be added to the CFRP to improve wetting of the carbon fibers by the matrix and bonding across the fiber–matrix interface both of which enhance load transfer, fillers are also employed in some polymer matrices to enhance dimensional stability and/or lower cost [1]. 

The functions of the polymer matrix in CFRP composites are to bind the fibers together and keep them fixed in desired geometric arrangement, transfer load to the fibers, protect fibers from chemical and mechanical damages, isolate fibers so that they are able to act independently resulting in slower crack propagation, optimize important performance indicators like ductility, and impact strength of the composite [98,99,100]. The desirable properties for a polymer matrix are: high toughness which should translate to a higher resistance to de-lamination, impact and a high failure strain and higher resin shear modulus which would result in better load transfer between fiber and resin and hence enhance compressive strength [101].

The functions of the carbon fiber in CFRP composites are to enhance the structural properties of the composite by carrying the load hence imparting strength to the composite, providing stiffness and enhanced thermal stability to the composite [99,102]. In order to perform these functions, the desirable properties for a reinforcing fiber in fiber reinforced polymer (FRP) composites are: excellent tensile properties in fiber direction which allows for high load transfer from the matrix in this direction, a high tenacity (i.e. strength, stress at maximum elongation) usually one or two orders of magnitude higher than that of the matrix, significantly lower maximum strain (ε_max_) of reinforcement fibers compared to the matrix which ensures that with good load transfer, composite failure occurs due to fiber breakage after undergoing its maximum elongation, and a high Young’s modulus which is a measure of stiffness [102]. Other desirable properties especially with respect to processing are a low variation of strength among fibers, high stability in the strength of the fibers during handling, and a close tolerance in the diameter and surface dimensions of the fibers [99].

Both thermosetting and thermoplastic polymers can be used as polymer matrices depending on the application. The thermosetting polymers used as polymer matrices in CFRPs are; epoxy [103,104,105], polyester [106,107], phenol formaldehyde [108,109], vinyl ester [110,111,112], bismaleimides and other thermoset polyimides [113,114,115,116,117,118], and cyanate ester [119,120,121,122,123,124]. The thermoplastic polymers employed as polymer matrices in CFRPs are; polyether ether ketone (PEEK) [3,6,125,126,127,128,129], polysulfone [130,131,132,133,134], polystyrene [135], polyphenylene sulfide [136,137,138], polycarbonate [139,140,141], polypropylene [142,143,144,145], nylon [146,147,148], thermoplastic polyimides [149,150,151], and thermoplastic elastomers such as butadiene-styrene diblock copolymer [152,153,154]. The choice of polymer matrix material for carbon-fiber reinforced polymer (CFRP) composites appear to be predominantly guided by the demands of its envisaged operating conditions, its mechanical properties and processability (Table 1). For instance the epoxy matrix is often employed in CFRP composites employed in the aeronautical industry while the polyether ether ketone (PEEK) polymer matrix is often employed in carbon fiber composites designed for medical applications.

For advanced high performance carbon fiber-polymer composite materials epoxy resin is the most commonly used polymer matrix [155]. The use of epoxy resins as thermosetting matrices of advanced composites employed in the electronics, aeronautical, and astronautical industries, is due to its possession of a good combination of interesting properties, such as good dimensional stability and chemical resistance, good stiffness and specific strength, and its strong adhesion to the embedded reinforcement phase [156]. The predominant use of epoxy as matrix material in advanced composites is due to its superior properties such as low shrinkage (2–3%), non-release of volatiles during curing, good processability, good storage stability that makes it easy to be employed in making prepregs, and a superior range of tensile modulus (2.7 to 5.5 GPa) coupled with good tensile strength (40 to 85 MPa) [101]. Table 1, reproduced with permission from the work of Mangalgiri [101], illustrates the important properties of epoxy that makes it a polymeric matrix of choice in advanced polymer composites.

Of all the polymer resins, cyanate ester is reported [119,157,158,159,160,161,162,163,164] to be a potential candidate thermosetting polymer matrix for aerospace applications alongside epoxy. This is based on its possession of the interesting combination of desirable properties of diverse resins, such as the workability of the epoxy resins, the thermal characteristics of bismaleimides, the heat and fire resistance of phenolic resins, and the fast curing that is characteristic of polyesters [119]. In addition cyanate ester appear to have a comparative advantage over epoxy resin with respect to moisture uptake, as it is reported [160,163,165,166,167] to exhibit low moisture uptake due to its non-hydrophilic cure mechanism [162], low dielectric constant and dissipation factor that is desirable in the electronics and aerospace industries [168].

## 3. The Nature of the Carbon–Epoxy Matrix Interface in Carbon-fiber reinforced polymers (CFRPs) and Its Effects on Composite Properties.

Besides the reinforcing carbon fibers and the polymeric matrix, another important factor that defines the properties of the composite is the interface/interphase between the carbon fibers and the polymeric matrix. This factor is of such importance that it is regarded as a third phase in reinforced polymer composites, with the strength of the composite strongly dependent on it [169]. The performance of a composite material is strongly dependent on the quality of the fiber–matrix interface [169], and the review of literature below indicates that a good part of the degradative processes in carbon-fiber reinforced polymers involves the interface. The interface region between fiber and resin is reported to be very complex, [170] and Hughes [171] had illustrated the structure of the interface region between carbon fiber-epoxy matrix (Figure 1).

The strength of the interface between the carbon fiber and the polymer matrix which ultimately and significantly determines the strength of the polymer composite can be traced back to the strength and nature of the intermolecular interactions between the molecules of the fiber and the polymer. Hence a proper appreciation of the nature of the carbon fiber–polymer interface from the molecular level upwards is vital to the design and development of high performance CFRPs with good combinations of high mechanical properties and high resistance to interfacial degradation (chemical stability) under cathodic polarization. Poor interfacial adhesion between the carbon fiber surfaces and polymer molecules has been attributed to the intrinsic hydrophobicity and chemical inertness of carbon [172]. To overcome this challenge and improve interfacial adhesion at the carbon fiber–polymer interface a lot of research efforts have been directed at the carbon surface [173], and although these reports are not reviewed herein the most important points from these reports are highlighted to set a molecular scale perspective into the interfacial strength of fiber reinforced polymers (FRPs). For a more detailed appreciation of this subject matter, the reviews by Jones [173], Tiwari and Bijwe [174], and Sharma et al. [172] are recommended. In studies on the modification of carbon surface and its effects on the chemistry of the carbon-polymer interface, changes in surface chemistry have been related to often macroscopically measured interfacial shear strength of the carbon fiber–polymer resin adhesion/bonding.

As a consequence fiber–matrix-adhesion (FMA) and inter-laminar shear strength (ILSS) or interfacial shear strength (IFSS) have emerged as important terms in the evaluation of the mechanical properties of the fiber-polymer matrix interface. By comparing the values of the inter-laminar shear strength, to the tensile strength of the reinforcing fibers, the polymer matrix alone, and the fiber-reinforced composite, along with the fractographs of the failed test specimens, important insights can be obtained on the quality of the designed composite with respect to mechanical (tensile) properties. In a “well designed” or “ideal” fiber reinforced composite in which a high tensile strength of the composite is the main design objective, failure is expected at the reinforcing fibers without interfacial delamination or fiber pullout. Such a scenario means that load was effectively transferred to the reinforcing fibers from the matrix, and that failure did not occur at the interface but at the reinforcing fibers at stresses around their failure stress. A very low inter-laminar shear strength is likely to result in failure by fiber pullout. Failure can theoretically occur by matrix-cracking if the matrix is stiffer than the reinforcing fiber (which is undesirable) and the interfacial adhesion is strong. It is necessary to highlight that a very strong interface is not always desirable. For certain applications in which a tough composite is required a very strong interface is not desirable so that fiber-pullout is can occur [173].

In order to improve interfacial adhesion of carbon fiber to the polymer matrix, carbon fibers are usually pre-treated. Most of the reported methods of pre-treating carbon fibers are oxidative [173] and include; acid treatment with oxidizing acids [175,176,177], oxidation in air or oxygen [173], electrochemical oxidation [178,179], plasma pre-treatment [173,180,181], and plasma (co)polymerization of resins onto carbon fiber surface [182,183], electro-deposition of polymers and co-polymers onto fiber surfaces [184,185], rare earth treatment [186,187,188], and recently carbon fiber surface modification by growth or introduction of nano-scale species (or multi-scaled carbon reinforcing fiber) [189,190,191,192,193,194,195,196]. 

The observed improvements in the fiber-polymer adhesion due to these treatments have been attributed to three mechanisms in the absence of a consensus [173]. The first mechanism is enhanced adsorption of the resin molecules onto surface complexes formed on carbon fiber surfaces as a result of the pre-treatment(s) with acidic complexes which are considered to be more effective [197,198]. The second mechanism is the emergence of a surface more suitable for polymer resin adsorption due to removal of surface contaminants from carbon fiber surface due to the pre-treatment. The third mechanism is the mechanical keying effect that enables the polymer resin to permeate pits and channels formed on the surface of carbon fibers that have undergone the oxidative pre-treatment [173].

To monitor the changes effected on carbon fiber surfaces due to these pre-treatment a number of analytical techniques have been employed which includes; titration methods [197], X-ray photoelectron spectroscopy (XPS) [176,180,199,200,201], ToF-SIMS [202], infrared spectroscopy [178,179], contact angle measurements, and electrokinetic measurements (streaming potential) [200], static secondary ion mass spectroscopy (SSIMS) [180], and inverse gas chromatographic (IGC) [200,203].

The pre-treatment of carbon fiber surfaces by oxidative methods which is reported [173] to improve fiber/resin adhesion have serious implications on the electrochemical reactivity and chemical stability of CFRP under cathodic polarization, as oxidized carbon surfaces have been reported by Taylor and Humffray [204] to result in oxygen reduction to OH^−^ instead of peroxide in alkaline media (pH > 10). This scenario is likely to lead to less degradation of the polymer matrix in CFRP composites containing carbon fibers that have been pre-treated in an oxygen environment prior to exposure to cathodic polarizations, as the presence of hydroxyl and peroxide ions are reported to enhance ring opening reactions in polymer matrices [205,206,207,208].

The strength of the fiber–matrix bond has been demonstrated to be a dominant factor in achieving maximum composite toughness [209,210,211,212]. However, in some instances optimum toughness can be achieved by a designed weak interface between the polymer matrix and the reinforcing phase [213]. In composites the integrity of the interface between the fiber and matrix (fiber–matrix-adhesion (FMA)) is usually monitored macroscopically via the inter-laminar shear strength (ILSS) [214,215,216,217,218]. The inter-laminar shear strength (ILSS) or interfacial shear strength (IFSS) often reported in N/mm^2^ characterizes the binding strength of fiber and matrix [212,219,220]. A shear stress between fiber and matrix exceeding the inter-laminar shear strength results in de-lamination [219]. The value of the ILSS between fiber and matrix is predominantly determined by two factors; the mechanical and the chemical interlock [102,221,222].

## 4. Efforts at Modification of CFRPs with Nanofillers and Possible Consequences for Conductivity and Galvanic Corrosion of Coupled Metals.

Recent trends in the evolution of modified carbon fiber based polymer composites appear to be quadro-pronged; towards nano-modification of fiber–matrix composite systems by use of carbon nanotubes and nano-fibers [155,223,224,225,226,227,228,229,230,231,232], the combined usage of nanometric reinforcing phases and nanometric matrix phases [191,223,224,225,226,227,228,229,230,231,232,233,234,235], the use of a composite matrix phase [236,237,238,239] or nano-composite polymer matrix phase [155,240,241,242,243] and nanometric modification of the interfacial region of the composite between the carbon fibers and the matrix [191,234,235,244,245,246,247,248], in attempts at improving the properties of the matrix phase. Many recently reported research efforts at improving the properties of carbon-fiber reinforced polymers by modification of the polymer matrix have been directed at using micro- and nano-sized particles as filler material for the epoxy [155] many of which have involved the use of carbon nanotubes/nanofibers (CNTs/CNFs). Carbon nanotubes/nanofibers (CNTs/CNFs) have emerged as popular materials for modification of CFRP composites for improved mechanical properties. Carbon nanofibers (CNFs) are hollow cylinders with a parallel and homogeneous alignment of nanoscopic graphene layers along the axis, with diameters typically in the range 50–500 nm and lengths of a few tens of microns which results in very high aspect ratios (length/diameter > 100) [232,249]. The popularity of carbon nanofibers (CNFs) as nanofillers in the modification of CFRP composites is premised on their very high mechanical and physical properties; (Young’s modulus ≈ 500 GPa, tensile strength ≈ 3 GPa, electrical conductivity ≈ 10^3^ S/cm, thermal conductivity ≈ 1900 W m^−1^ K^−1^) [249]. The use of carbon nanotubes to reinforce polymer composites is predicted to potentially result in nanotube-reinforced polymers possessing 20% of the theoretical properties of carbon nanotubes [164].

The use of carbon nanotubes (CNTs) as fillers for modifying CFRPs is due to their (smaller) nano-dimensions compared to carbon nano-fibers (CNFs) and their excellent mechanical properties that qualifies them as ideal filler materials [250]. With diameters in the range of 1 to 100 nm and lengths in the millimeter range [251], densities as low as 1.3 g/cm^3^, and Young’s moduli with values greater than 1 TPa (which is superior to that of all types of carbon fibers [252]) and uniquely high strengths of which the highest measured value for a carbon nanotube was 63 GPa [253] (which is an order of magnitude stronger than that of high strength carbon fibers [254]) addition of carbon nanotubes to CFRP is bound to result in improvements in mechanical properties, and accounts for the superior enhancements obtained with CNTs compared to results from the use of CNFs.

Some of these strategies have produced measured (but often mixed) successes at improving different properties of the epoxy matrix phase and the entire polymer composite, with enhancement of certain properties but reduction in other mechanical properties. Employing carbon nanofibers as fillers, Zhou et al. [155] reported 11% and 22.3% increase in the tensile and flexural strengths respectively of CFRP composite in which the epoxy matrix was filled with 2% carbon nanofibers compared with same composite without addition of carbon nanofibers to the epoxy. In another report, Zhou et al. [241] studied the effect of introducing multi-walled carbon nanotubes (MWCNTs) into the epoxy matrix on the mechanical properties of the epoxy-carbon nanotube composites and reported the highest improvements in both strength and fracture toughness at 0.3 wt % CNT loading. Incorporating the nano-phased matrix filled with 0.3 wt % MWCNTs to woven carbon fibers to obtain an epoxy-MWCNT-carbon fiber composite they reported improvements in the glass transition temperature, decomposition temperature, and flexural strengths of the epoxy-MWCNT-carbon fiber composite. 

Gojny et al. [224] did a comparative study of the effect of different carbon nanotubes (single-wall CNTs (SWCNT), double-wall CNTs (DWCNTs), and multi-wall CNTs (MWCNTs)), and amino-functionalization of DWCNTs and MWCNTs on the mechanical properties of epoxy-based nanocomposites and reported generally superior improvements in the mechanical properties (Young’s modulus, ultimate tensile strength, and fracture toughness) with amino-functionalized DWCNTs at higher (0.3 and 0.5 wt %) nano-filler content. The superior improvements in mechanical properties with the amino-functionalized CNTs was attributed to improved dispersibility due to increase in the surface polarity of the functionalized CNTs which induces their incorporation into the matrix network. Smaller improvements with SWCNTs was attributed to their very high specific surface areas (of up to 1300 m^2^/g [255]) which made dispersion much more difficult. 

With respect to the approach of composite modification by modifying the interface between fiber and matrix by deliberate placement of fibers in the interface region (Figure 1), besides the weight fraction of fibers introduced in this region other important variables are present. These other variables might include the fibers’ distribution pattern in the interfacial region, their orientation with respect to the fiber, and their length profile [226] which we postulate would be advantageous, if their lengths are longer than the width of the interfacial region. Wicks et al. [191] studied the effects of the growth of carbon nanotubes onto the carbon fibers (thus modifying the interface between fiber and matrix) and into the matrix (thus modifying the matrix) on the mechanical properties of the carbon fiber reinforced composite and reported that the growth of aligned CNTs onto the carbon fiber surface bridged the ply interfaces resulting in enhancement of both the initiation and the steady-state toughness, with improvements of up to 76% in steady state (with actual values increased by more than 1.5 kJ/m^2^). They reported marked differences in the failure mechanism, in which shear-out failure (matrix dominated) is predominant without CNTs, while tensile fracture (fiber dominated) is more prevalent on introduction of the CNTs.

For a more in-depth appreciation of the impressive enhancements of mechanical properties of polymers using carbon nanotubes as filler materials the review by Coleman et al. [250] is recommended, as the present work is more focused on the effects such attempts at modification of the carbon fiber polymer composite (especially the polymer matrix with conductive phases) might have on the electrochemical activity of the composite under cathodic polarization(s) consistent with that envisaged on its galvanic coupling to metals. 

Besides changes in mechanical properties, modification of carbon fiber - epoxy composites exert significant influences on the electrical properties of the composite which has implications on its electrochemical behavior on galvanic coupling with metals as a component of multi-material assemblies. Several reports on the (changes in the) electrical properties of CNF modified epoxy composites are found in the literature [232,256,257,258,259,260,261,262]. One of the ways in which the conductivity of a composite is enhanced by the introduction of conductive fillers like carbon nanofibers is by percolation. Percolation occurs when conductive fibers added into a non-conducting matrix material form conductive paths within the material, referred to as a so-called percolating network. The filler (e.g., CNFs or CNTs) concentration at which the percolating network just forms as filler concentration is increased is regarded as the percolation threshold and is marked by a sudden increase in the conductivity of the composite [263].

Cipriano et al. [264] studied the effect of addition of multi-walled carbon nanotubes (MWCNTs) or carbon nanofibers (CNFs) to polymeric melts comprised of polystyrene (PS) and different annealing temperatures and times on the electrical conductivities of the resultant (PS/MWCNT and PS/CNF) nanocomposites and reported higher electrical conductivity values with carbon nanotubes compared to carbon nanofibers of similar weight fraction. For PS/MWCNT composites (at 1, 2, and 4 wt % MWCNTs) the conductivity values extracted from their plots where in the range of 1 × 10^−8^ S/m (the lower detection limit of their test system) to up to 1 S/m after annealing at temperatures 170 ^o^C. In the case of PS/CNF composites (at 3, 5, 7, 10, and 15 wt % CNFs) the conductivity values extracted from their plots where in the range of 1 × 10^−8^ S/m (the lower detection limit of their test system) to values in the range of 10^−3^ S/m without annealing, but rises at higher CNF weight fractions (≥ 10 wt %) and after annealing at temperatures ≥ 170 °C to values close to 1 S/m. They attributed the higher conductivity values measured in composites containing MWCNTs to the CNFs lower intrinsic conductivity and lower aspect ratio compared to MWCNTs [264,265]. 

Pötschke et al. [266] studied the rheological and dielectric properties of polycarbonate (PC)-MWCNT nanocomposites with 23 different contents of MWCNTs ranging from 0.1 to 12.5 wt % MWCNTs and reported a temperature dependence of the electrical percolation threshold, (about 1.0 wt % MWCNTs at room temperature), and significant (greater than 7 decades) changes in the low-frequency conductivities of the different nanocomposites in the studied (0.1 to 12.5 wt %) MWCNTs concentration range. Extracts of electrical conductivity at 1 Hz from their plots of the frequency dependence of the conductivities of composites with different MWCNT content indicates that at 0.75 wt % MWCNT the conductivity is ≈ 10^−14^ S/cm, ≈ 10^−10^ S/cm at 1 wt % MWCNT, ≈ 10^−6^ S/cm at 1.125 wt % MWCNT, ≈10^−4^ S/cm at 1.375 wt % MWCNT, ≈ 10^−2^ S/cm at 2 wt % MWCNT, and > 10^−2^ S/cm at ≥ 2.5 wt % MWCNT. By employing either shearing or the application of an electrical field to control the re-agglomeration of the CNTs, ultralow percolation thresholds as low as 0.0025 wt % of multi-wall carbon nanotubes (MWCNTs) in carbon-nanotube-epoxy composites have been achieved [267]. demonstrating the feasibility of improving the electrical properties of polymers with ultralow CNT contents.

Alig et al. [268] studied the electrical properties of polycarbonate nanocomposite mix with 0.6 vol % MWCNT (corresponding to 0.875 wt % MWCNT) in a rheometer, and reported a time dependent recovery of the DC-conductivity in the rest time after a transient shear deformation. The observed tremendous increase of the DC-conductivity with time after shearing was attributed to the re-organization of the conducting MWCNT network, which was destroyed by the shear, while the reformation kinetics of re-organization of the MWCNT-clusters was considered as a co-operative cluster–cluster aggregation coupled to electrical percolation of conductive MWCNTs. From their plot of DC conductivity versus MWCNT content they observed that the electrical percolation appears at 0.68 vol % (equivalent to 0.875 wt % MWCNT), while the melt sample containing containing 0.6 vol % of MWCNTs (below percolation) showed a conductivity of about 10^–3^ S/cm, after annealing at T = 230 °C which was concluded to be indicative of a change from a non-percolated state into a percolated one. From the analysis of the transmission electron microscopy (TEM) images of PC/MWCNT composites after annealing at T = 280 °C and 300 °C respectively, they concluded that annealing of the sample shown at T = 300 °C results in the nanotube aggregation process and a consequent increase of the composite conductivity, and postulated that nanotube aggregation is the key process accounting for the changes in electrical conductivity in polymer-MWCNT melts. By measuring the electrical conductivity at different combinations of shearing rates and annealing temperatures, they observed that during annealing at T = 230 °C the DC-conductivity of the as-produced non-conductive PC/MWCNT composite increased by several orders of magnitude. Arguing that such marked increase in electrical conductivity of the polymer matrix (from 10^–15^ S/cm to 10^–1^ S/cm) can only be partially attributed to the conductivity increase of the polymer matrix (from 10^–15^ S/cm to 10^–1^ S/cm) with the increase in temperature, they attributed the predominant cause of the increased conductivity to the creation of conductive nanotube pathways in the polymer matrix through a self-organization processes. This position is supported by the observance of a marked drop in conductivity down to values of ≈10^–10^ S/cm after application of a short mechanical deformation to the hitherto more conductive sample and a complete recovery of the DC-conductivity to its starting value after a rest time. The recovery of the electrical conductivity in the melt state after a mechanical deformation was attributed by the authors [268] to the rebuilding of the conductive nanotube network. These observations by Alig et al. [268] have important implications for the use of modified carbon reinforced polymers in applications in which increase in temperature and presence of some stress are feasible; as the combination of these two factors have been demonstrated to be capable of exerting marked changes in the electrical conductivity of the modified polymer matrix. An example of such application is the use of carbon reinforced polymers in weight optimized multi-material combinations often employed in the aeronautical industry, in which the possible changes in the electrical conductivity of the polymer matrix can affect the anodic dissolution of active metals galvanically coupled to carbon reinforced polymers in the presence of an electrolyte.

Pötschke et al. in another report [269] studied the effect of two different processing methods on the dispersion of carbon nanotubes into thermoplastic polymers melts as the enhancement of mechanical properties and electrical conductivity in polymer composites is strongly dependent on the homogenous dispersion of the nanotubes in the polymer matrix. They reported that addition of carbon nanotubes significantly changed the stress-strain behavior of the composites, with increased modulus and stress while elongation is reduced, particularly at CNT concentrations above the percolation concentration. They reported that irrespective of processing method and polymer matrix type, in all the cases, polymer-CNT composites with contents greater than 1.5 wt % MWCNT can be regarded as electrically conductive (with volume resistivity < 10^4^ Ohm-cm) which is equivalent to a conductivity of 10^–4^ S/cm, and attributed relatively low percolation thresholds observed in some instances to good distribution and dispersion of carbon nanotubes within the PC matrices. 

In an earlier report Pötschke et al. [270] studied the complex permittivity and the related AC conductivity of polymer composites of polycarbonate (PC) filled with different amounts (0.5 and 5 wt %) of multi-walled carbon nanotubes (MWCNT) and reported that the differences in the dispersion of the MWCNT in the PC matrix could be detected in the complex permittivity and AC conductivity spectra. Their report shows that the electrical conductivity of the un-filled polycarbonate polymer increased from 10^–16^ S/cm to > 10^–2^ S/cm with 5 wt % of multi-walled carbon nanotubes (MWCNT).

Lozano et al. [264] studied the effects of the mixing rheology on the conductive properties of CNF-polypropylene composites in the concentration range of 0 to 40 wt % CNFs and reported electrical conductivity values ranging from 1 × 10^−17^ in unfilled polymer to ≈ 10^−7^ S/cm in CNF filled polymers.

Table 2 summarizes some of the reported values for electrical conductivity in S/cm of modified carbon–polymer (epoxy) composites, together with the modifying fillers used and their concentrations or range of concentrations. In some instances, reported electrical conductivity values in other units have been converted to S/cm and/or reported values of electrical resistivity converted to conductivity for clarity and ease in comparison of reported data from different sources.

Bal’s [232] report of 3–6 orders of magnitude increase in the electrical conductivity of insulating epoxy after its infusion with carbon nanofibers, supports our position that these efforts at nano-modification of carbon fiber reinforced composites is capable of enhancing its electrochemical activity with deleterious implications for multi-material corrosion of multi-material combinations containing these nano-modified CFRP. 

The marked increase in the conductivity of carbon nanofiber filled epoxy at low concentration (0.1 to 1% CNFs content) by Bal [232] which is much lower than the percolation limit for most polymers filled with carbon black which is reported [271] to be in the range of 5% to 20% carbon filler content excludes percolation as the cause of the enhanced conductivity of the nano-filled composites. In the light of the fact that at such very low nano-filler concentrations (0.1% to 1% CNF content) employed by Bal [232] the nanofibers are unlikely to touch each other directly to form a continuous electrical path, the enhanced electrical conductivity of the nano-fiber filled nanocomposites was attributed to reduction in the distances between nanofibers to values lower than the hopping distances of the conducting electrons [232,264]. On the basis of the conductivities values reported for CNF-epoxy nanocomposites by both Cipiriano et al. [264] (≈ 10^−6^ S/cm with ≥ 7% of CNF), and Bal [232] (2 × 10^−6^ S/cm to 4 × 10^−3^ S/cm with 0.5–1% of CNF) vis-a-vis the conventional classification of materials with electrical conductivities < 10^−6^ S/cm as insulators, materials with electrical conductivities between 10^−6^ S/cm to 10^−2^ S/cm as semiconductors and those with electrical conductivities above 10^−2^ S/cm as conductors [272,273,274], it is obvious that the modification of the epoxy matrix with carbon nanofibers is capable of shifting the electrical properties of the epoxy matrix from values that are consistent with insulators (in unmodified epoxy) to values consistent with semiconductors even at very low carbon nanofiber/nanotube content in nano-filled epoxy matrix filled with carbon nanofibers. 

The effect of the introduction of CNTs and CNFs into CFRP as nano-fillers on the galvanic corrosion of technologically relevant metals/alloys coupled to CFRP is an area that require detailed study. However, there are just few reports in literature on galvanic corrosion of metals coupled to nano-filler modified fiber reinforced polymers [275,276,277] Ireland et al. [275] investigated galvanic corrosion between aluminum 7075 and glass fiber reinforced polymer (GRFP) composites modified with carbon nanotubes and reported statistically significant increase (approximate doubling) of corrosion rate and mass loss rate on coupling MWCNT/GFRP samples with aluminum 7075 compared to baseline GFRP samples. It is important to note that in the MWCNT/GFRP - AA7075 galvanic couple studied by Ireland et al. [275], the reinforcing fiber (glass fiber) in the GFRP is not electrically conductive in contrast to CFRP in which the reinforcing fiber (carbon fiber) is electrically conductive. Baltzis et al. [276] investigated the performance of austenitic stainless steel 304 (SS304) adhesively bonded to neat epoxy, CNT-modified epoxy and carbon fiber (CF)-reinforced epoxy (CFRRP), and reported that though the incorporation of CNTs increased the galvanic effects, it also retarded uniform corrosion and localized corrosion as the modified adhesives prevented the electrolyte from reaching the substrate. Arronche et al. [277] studied galvanic corrosion between AISI 1018 carbon steel coupled to CFRPs modified with multi-walled carbon nanotubes and reported that the addition of MWCNTs do not have a statistically significant effect on the corrosion and mass loss rates compared to unmodified CFRP. Comparing their result [277] which is at variance with other works [275,276] that reported “CNT-induced” increase in galvanic corrosion of metals coupled to fiber reinforced polymer composites, they attributed their non-observance of an increase in metallic (AISI 1018 carbon steel) corrosion rates on galvanic coupling to MWCNTs modified CFRP to the already conductive nature of carbon fiber reinforcement with respect to the CNT fillers. This attribution in our opinion is most probably incorrect, because addition of CNTs above its percolation threshold in the polymer matrix is bound to increase the conductivity of the matrix phase of the composite so that the entire CFRP composite surface becomes conductive. Ideally, such increase in the conductive area of CFRP surface on addition of CNTs is expected to lead to an increase in the cathodic area of MWCNTs modified CFRP - AISI 1018 carbon steel galvanic couple, which should ordinarily support increased anodic dissolution of the coupled carbon steel. However, from work carried out in our laboratory [278] which studied galvanic corrosion and inhibition in Al-CFRP, Cu-CFRP, Zn-CFRP, and Fe-CFRP galvanic couples and Figure 2 of this work, it is obvious that unlike CFRP coupled to Al, Zn, and Mg, at the cathodic potentials CFRP is polarized (around −400 mV_SCE_) cathodic reactions on CFRP surface might not yet be under diffusion control. Hence increase in the conductive surface area of the CFRP composite by modification with the MWCNTs might be less likely to translate to enough increase in cathodic activity that can support a statistically significant increase in anodic dissolution of Fe or carbon steel. This can explain the contrasting trend observed by Arronche et al. [277] for galvanic corrosion between AISI 1018 carbon steel coupled to CFRPs modified with multi-walled carbon nanotubes, as all reports of “CNT-induced” galvanic corrosion of metals coupled to CNT-modified CFRPs involved metals/alloys that are able to polarize CFRP to potentials at which cathodic processes on CFRP surface are under diffusion control. 

## 5. The Driving Force for Galvanically Stimulated Degradation of Carbon Fiber Reinforced Polymer Composites

The galvanic potential of the carbon-based materials is normally more positive than that of most of the structurally relevant metallic materials. Therefore on electrical contact with metals in the presence of an electrolyte, CFRP is cathodically polarized. The electro-catalytic activity of carbon surfaces towards oxygen reduction reactions are reasonably high while the hydrogen evolution reaction is strongly suppressed leading to a wide electrochemical window of water. Thus the cathodic processes occurring on the carbon surface, when galvanically coupled to most of the metals and alloys, are associated with reduction of oxygen dissolved in the electrolytes at different relevant technological conditions. Under naturally aerated conditions the cathodic processes in the established galvanic systems normally occur under oxygen diffusion control for many metals (Figure 2). Since the cathodic current densities on CFRP are reasonably high, CFRP is capable of supporting significant anodic dissolution of metals in electrical contact with it.

## 6. Mechanism of Oxygen Reduction on Carbon Electrodes

From Figure 2 above, it is obvious that the major cathodic reaction on the surface of carbon and carbon reinforced polymers galvanically coupled to most metals is oxygen reduction. Hence an appreciation of the mechanism of oxygen reduction on carbon electrodes is vital to understanding multi-material corrosion of metals galvanically coupled carbon reinforced polymers in multi-material assemblies. Oxygen reduction on carbon and graphite electrodes reportedly proceeds mainly through the peroxide pathway [279,280,281]. The general mechanism of oxygen reduction on ordinary graphitic electrode in the absence of added catalysts involves two successive oxygen reduction processes (Equations (1) and (2)) [279,280,282,283];
(1)$$O_{2}+H_{2}O+2e^{-}\;⇌\;HO_{2}^{-}+OH^{-}\;\left(E^{o}\;\approx \;-309\;mV\;vs.\;SCE\right)$$

The reaction of Equation (1) is either followed by another 2-electron (electrochemical) reduction of the hydroperoxyl (HO_2_^–^) ion to hydroxyl ion (OH^–^), expressed in Equation (2);
(2)$$HO_{2}^{-}+H_{2}O+2e^{-}\;⇌\;3OH^{-}\;\left(E^{o}\;\approx \;-1111\;mV\;vs.SCE\right)$$
or by a fast chemical decomposition of the hydroperoxyl (HO_2_^–^) ion that leads to regenerative production of oxygen (Equation (3));
(3)$$HO_{2}^{-}\;⇌\;\frac{1}{2}\;O_{2}+OH^{-}$$

The rate of the chemical decomposition of the hydroperoxyl (HO_2_^–^) ion (Equation (3)) relative to its consumption rate in the 2-electron electrochemical reduction process (Equation (2)) is a factor in detection of a second peak in cyclic voltammograms attributable to the electrochemical reduction of the hydroperoxyl (HO_2_^−^) ion to hydroxyl ion (OH^–^) [283].

In spite of the fact that a full-scale mechanistic analysis of the oxygen reduction reaction (ORR) is inexpedient [283], there appears to be a general consensus [280,284,285,286] that the process starts with or involves surface adsorption of oxygen on metal-free carbon surfaces (Equation (4)), followed by an initial electron-transfer step leading to O_2_ reduction to superoxide (Equation (5)), a protonation step leading to formation of hydroperoxide radical (Equation (6)), and its subsequent reduction to hydroperoxide (Equation (7)) [283]. This treatment is equivalent to considering the first 2-electron reaction (Equation (2)) to take place as a four-step 2-electron process expressed in (Equations (4)–(7)) below.

An adsorption step:(4)$$O_{2}\;\to \;O_{2\;\left(ads\right)}$$

An electron-transfer step which is reportedly [286,287] the likely rate determining step at pH < 10 while surface migration of O_2_^−^ions to active sites on the electrode surface is reported to be the rate determining step at pH > 10 [204].
(5)$$O_{2\;\left(ads\right)}\;⇌\;O_{2\;\left(ads\right)}^{*-}$$

A proton-transfer step
(6)$$O_{2\;\left(ads\right)}^{*-}+H_{2}O\;⇌\;HO_{2\;}^{*-}+OH^{-}$$
and then another electron transfer process
(7)$$HO_{2\;\left(ads\right)}^{*}+e^{-}\;⇌\;HO_{2\;\left(aq\right)}^{-}$$

## 7. The Surface Chemistry of Carbon and Its Effects on Cathodic Processes on CFRPs.

The ability of the carbon fibers in CFRP to support cathodic reactions is intimately linked to the surface chemistry of carbon which can be complex and dynamic, as carbon is capable of forming “surface oxides” of oxygen containing and even nitrogen containing functional groups (Figure 3) [288,289,290,291] which affect surface reactivity. These “surface oxides” have been classified as basic [292] or acidic [293] depending on the acidity of the functional groups present on the surface. The introduction of oxidizing agents to carbon materials under wet or dry conditions can lead to the formation of either of three types of oxygen containing surface groups; acidic, basic, and neutral [294,295,296]. Gas phase oxidation of the carbon is reported to lead mainly to increase in the concentration of hydroxyl and carbonyl surface groups, while oxidations in an electrolyte predominantly increases the concentration of carboxylic acids [297,298]. The application of polarization either by external polarization or by galvanic coupling to metals is capable of changing the type of surface groups [299] on these carbon fiber surfaces. Anodic oxidation of graphitic materials in aqueous solutions reportedly leads to formation of surface oxides predominantly composed of carboxylic and phenolic groups [300,301,302]. Cathodic polarizaton of carbon surfaces containing “surface oxides” is likely to result in some reduction of these oxides [303,304,305,306]. Basova et al. [307] studied how the surface modification of carbon fiber composites by oxidation–reduction cycles affected their electrochemical behavior towards oxygen evolution and concluded that the presence of acidic functional groups (such as –COH, –COOH) on the carbon fiber surface enhances the electrochemical interaction between the oxidized carbon fiber surface and electrolyte solution, while the presence of phenolic, hydroxyl, and quinone groups inhibit this interaction. 

The specific capacitance electrochemical double layer on carbon surface is reportedly structure sensitive [308] and increases with formation of carboxyl groups on the surface during anodic oxidation [309,310,311]. Without special pre-treatments almost all carbon surfaces are prone to reactions with oxygen and water even without polarization, resulting in the formation of oxygen-containing functional groups on the carbon surfaces. Hence the exposed surfaces of carbon fibers in the carbon-fiber reinforced polymers in quiescent blank solutions, are most probably oxygen terminated. This can result in formation of “surface oxides” or oxygen-containing functional groups on the carbon fiber surfaces on interaction with air and moisture during polishing, and also on immersion in aqueous media whilst galvanically coupled to metals which leads to a high pH environment near the CFRP. These surface oxide species can have effects on the general electrochemical behavior of the carbon fibers, and on specific processes such as adsorption, electron transfer kinetics, and electrocatalysis. For instance, negative surface charge due to carboxylates as surface oxides on carbon, have been reported to exert marked electrochemical effects on adsorption and electron transfer rates [312,313,314].

Though the presence of these surface oxides are almost inevitable and undesirable, it is possible in some situations to manipulate them for beneficial electrochemical performance [315]. The application of a high current density through carbon fibers in an electrolyte solution is reported to have extraordinary effects on the fiber morphology, by the generation of a very high surface area with an apparent capacitance up to 4000 µF cm^−2^, which is more than two orders of magnitude higher than the typical values for glassy carbon and carbon fibers [315,316,317,318,319]. Generally the double layer capacitance of carbon materials is reported to be about 20 μF cm^−2^ or less [320]. 

In another dimension to the surface chemistry of carbon, graphitic materials are regarded to be composed of at least two well defined surface sites; the edge plane sites perceived to be the active sites for electrochemical reactions, and the basal plane sites regarded as relatively less active electrochemically as improvements in electrochemical behavior have been reported by increasing the edge sites [321]. For a single layer graphene sheet much larger specific capacitance, faster electron transfer rate, and stronger electrocatalytic activity have been reported for the graphene edge compared to the basal plane [322]. Recently direct evidence of the greater electrochemical activity of graphite edge sites towards the oxygen reduction reaction has been demonstrated (Figure 4) [323].

Marked differences have been reported in the double layer capacitances of the basal and edge sites of graphite with capacitance values of 3–16 μF cm^−2^ reported for the basal sites, and 50–70 μF cm^−2^ for the edge sites respectively in 0.9M NaF solutions [325]. Chu and Kinoshita [321] have suggested that such marked differences in the double layer capacitance of basal and edge plane sites can be exploited as a convenient diagnostic tool to monitor changes to the carbon surface, and anticipated that modifications to the carbon surface that increase the concentration of exposed edge sites on the basal plane surface will result in higher values of double layer capacitance. Conversely, we anticipate that surface modifications to the carbon surface that result in decrease in the concentration of exposed edge sites on the basal plane surface is likely to result in lower double layer capacitances.

In addition, the type of surface functional group on the carbon surface can affect the kinetics of oxygen reduction on carbon electrodes. Rao et al. [326] reported best ORR performance from vertically aligned carbon nanotubes (VA-CNTs) with a nitrogen concentration of 8.4 atom % and ascribed it to a greater number of pyridinic-type nitrogen sites. Buan et al. [327] worked with CVD synthesized nitrogen-doped carbon nanofibres supported on expanded graphite, and reported that N doped CNFs (N-CNFs) revealed differences in the microstructure, nitrogen content and nitrogen composition depending on the growth catalyst used during synthesis and that N-CNFs grown from Fe demonstrated high ORR onset potentials and selectivity towards the 4-electron pathway in both acidic and alkaline electrolytes and attributed this to a high pyridinic nitrogen content.Maldonado and Stevenson [283], used electrochemical methods to study the influence of nitrogen doping on ORR on carbon nanofiber (CNF) electrodes in aqueous KNO_3_ solutions at neutral to basic pH, and reported differences in the mechanism of oxygen reduction. According to their report ORR proceeds by the peroxide pathway via two successive two-electron reductions on undoped carbon surface, but on N-doped CNF electrodes, proceeds as a catalytic regenerative process in which the intermediate hydroperoxide (HO_2_^−^) is chemically decomposed to regenerate oxygen, with a spectacular ≈ 100 fold enhancement of hydroperoxide decomposition process. This observed enhanced activity was attributed to the presence of edge plane defects and nitrogen functionalities within the carbon nanofiber (CNF) structure. Working with un-doped and nitrogen doped carbon nanotubes, Wiggins-Camacho and Stevenson [328] established a positive correlation between nitrogen content and ORR activities in which the ORR at undoped CNTs proceeds via two successive two-electron processes with hydroperoxide (HO_2_^–^) as the intermediate, but proceeds via a “pseudo” four-electron pathway involving a catalytic regenerative process in which hydroperoxide is chemically decomposed to form hydroxide (OH^–^) and molecular oxygen (O_2_) at nitrogen-doped carbon nanotubes, and reported an even better >1000-fold enhancement for hydroperoxide disproportionation. By analysis of observed charge-transfer coefficients, α_obs_ in an earlier article [283] they had concluded that for N-doped CNFs the charge-transfer coefficient, was consistent with a strongly adsorptive pathway while that of the un-doped CNFs being invariant to solution pH, was suggestive of the absence of strongly adsorbed O_2_ at the undoped CNFs in the studied pH range. Furthermore, they [283] concluded that the absence of the second reduction peak in the N-doped CNF electrode in contrast to its presence in the un-doped CNF electrode was suggestive of faster rate of chemical decomposition of the electrochemically generated hydroperoxide to oxygen compared to the rate at which it is electrochemically reduced to OH^−^.

In energy research pertaining to fuel cells and batteries, iron, platinum, and nitrogen are incorporated into carbon electrode materials to provide catalytic sites for electrocatalysis of O_2_ reduction to both hydrogen peroxide and water [283,315,327,329,330]. This is interesting with respect to the use of carbon-fiber reinforced polymers together with metals in various forms such as laminates [331] or in processes for joining metals to carbon-fiber reinforced polymers [89,332,333,334,335] in which contamination of carbon fiber surface with metal particles might create the possibility of enhanced cathodic activity on CFRP components of such hybrid structures, and hence accelerated dissolution of metallic components of such structures. The ability of the electrically conductive carbon fiber component of CFRP composites to support cathodic reactions on galvanic coupling to metals in multi-material assemblies is capable of affecting the structural integrity of such hybrid structures in two major ways; (a) by accelerating dissolution of metallic components of such structures, and (b) by accelerated degradation of the carbon-fiber reinforced polymers either in the form of interfacial damage and/or degradation of the polymer matrix by aggressive products from cathodic processes such as peroxide species. Since the main cathodic reaction on CFRP is oxygen reduction by the carbon fibers, oxygen reduction on carbon electrodes is discussed before CFRP degradation.

## 8. Carbon-Fiber Reinforced Polymers (CFRPs) Degradation under Cathodic Polarization.

Uvarov et al. [336] studied the effect of electrochemical treatment (cathodic, anodic, and a combination of these) on the morphology and surface chemical composition of carbon fiber materials and reported that whereas anodic polarization resulted in a marked decrease in conductivity with significant weight gain, a single cathodic polarization had no effect on specific conductivity but resulted in slight reduction in weight due to ´degradation of fiber surface and the attendant loss of material.

Tucker et al. [337] studied the behavior of graphite/epoxy composites galvanically coupled to a range of metals in seawater and reported degradation in form of blisters on the composite when active corrosion of aluminum occurred which is not observed without galvanic coupling and hence linked to enhanced cathodic chemical reactions associated with galvanic corrosion of the metal.

Sloan [338] carried out long-term environmental exposure (one year) to study the degradation of graphite/epoxy composites in 5% solutions of H_2_O_2_ at pH values of 6, 10 and 12 using SEM and FT-IR analysis, and reported based on the spectroscopic results that hydrogen peroxide electrophilically attacks the secondary amines in the cured epoxy structure with SEM images showing general decomposition of the matrix, but found no evidence of attack by the high pH solutions. The graphite/epoxy composites employed in this study had a matrix composed of tetraglycidyl diaminodiphenyl methane (TGDDM) epoxy systems cross-linked with the use of the bi-functional cross-linking agent diaminodiphenylsulfone (DDS). The authors [338] proposed a mechanism in which the hydrogen peroxide would first form the amine hydroxide, which under mildly acidic conditions undergoes hydrolysis (dilute solutions of H_2_O_2_ being mildly acidic [339], followed by an H_2_O addition reaction and finally the formation of an aldehyde. Sloan and Talbot [340] in a later study on the evolution of perhydroxyl ions on graphite/epoxy cathodes demonstrated clearly that both hydroxyl and perhydroxyl ions are evolved during cathodic polarization of graphite/epoxy. They concluded that the earlier mechanism explaining the degradation of polymeric matrix materials in carbon fiber reinforced polymer (CFRP) composites, in which damage is attributed to nucleophilic attack of hydrolyzable polymer linkages (e.g., esters, imides, etc.) by hydroxyl ions (OH^−^) evolved at the cathode, though seemingly plausible for damage to the hydrolyzable polymers this postulation, is insufficient to explain the damage observed in non-hydrolyzable polymers. On the basis of their experimental observations they proposed a new mechanism accounting for the degradation of both hydrolyzable and non-hydrolyzable polymers in which degradation is attributed to the build-up of deleterious concentrations of highly reactive perhydroxyl ions (HO_2_^−^) in occluded regions of the composite with high pH solutions, in which the evolved perhydroxyl ion is both stable and highly reactive as these are strongly dependent on solution pH.

Pauly et al. [341] in the section of their work on effect of cathodic polarization on the durability of pultruded graphite/epoxy composites in aerated 0.6 M NaCl concluded that the high pH environment generated during the oxygen reduction reaction was a necessary but not a sufficient condition for composite degradation because un-polarized specimens exposed to a pH 13 environment exhibited no degradation. They [341] postulated that cathodic polarization of a magnitude achievable by galvanic coupling to steel or aluminum is needed and that degradation of the fiber/matrix interface is most probably due to the intermediates (peroxide (HO_2_^−^) and long-lived super oxide species) generated during the oxygen reduction reaction and not the OH^−^ that are final reaction products. They [341] emphasized a damage mechanism for CFRP degradation under cathodic polarization proposed by earlier authors [205,342,343] based on the premise that the two-electron pathway capable of producing deleterious intermediate species that are able to interact with certain polymer bonds in the matrix and in a lot of cases degradation is the favored mechanism for oxygen reduction on carbon. In this proposed mechanism, oxygen is reduced to hydroxyl ions at the surface of the graphite fibers leading to the generation of local areas of high pH which in occluded areas near the fibers is even higher promoting the build-up of peroxide radicals. These reactive peroxide radicals then attack and degrade the polymer matrix in close proximity to the graphite fiber, leading to a break-down of the fiber/matrix interface; a process which continues until solution ingress occurs along the fiber causing an increase in the active surface area of the electrode and inducing a porous electrode response. In addition to the mechanism summarized above, Taylor et al. [343] postulated from their results that although hydroxyl ions were regarded to be the damaging species, the superoxide radicals produced during homogeneous peroxide decomposition can be the major deleterious species owing to their ability to react with labile hydrogens on the polymer surface to produce the even more highly reactive hydroperoxyl radical (HOO*) which being hydrocarbon soluble can readily diffuse through the polymer matrix in which its further reaction with labile hydrogen produces peroxide [344]. Damage to the polymer matrix due to cathodic polarization is reportedly marked in polymer composites with polymer matrices that are capable of ring opening reactions in the presence of hydroxyl and peroxide ions [205,206,207,208].

Tang et al. [345] studied the degradation of carbon/vinyl ester composites under cathodic polarization in seawater and reported that both the flexural modulus and flexural strength decreased with the increase in cathodic polarization, with the scanning electron micrographs showing that under cathodic polarization the polymer matrix was locally detached and some carbon fibers released from the specimens.

## 9. Chemical Stability/Degradation of the Polymer Matrix 

Since CFRPs are produced with various polymer matrices to achieve diverse design objectives, the chemical stability/degradation of some of these polymer matrices is reviewed. With respect to the degradation of these polymer matrices in aqueous media and under cathodic polarizations in the range expected on galvanic coupling with metals, the vital factors with respect to the polymers need to be highlighted. These vital factors are most likely to be; the polymer molecular weight or molecular weight distributions, the types of “linkages” and end-groups present, and the possibility of increasing the cross-link density in thermosetting resins by reactive species (O_2_^−^, O_2_^2−^, OH^−^, HO_2_^−^) produced from cathodic process which can lead to embrittlement. Few reports have been made on polymer matrix degradation under these conditions to aid a comprehensive review of the degradation of all the types of polymer matrices employed in advanced composites. However the most relevant reports on epoxy and bismaleimide are reviewed. The hydrolysis of polyimides at high pH has been reported [346,347]. 

Woo et al. [208] studied the degradation of toughened and untoughened epoxy and bismaleimide (BMI) carbon fiber reinforced composites in terms of their matrix chemical stability in galvanic reactions between metals and composites and reported the absence of degradation in any of the epoxy composites (toughened or untoughened) due to galvanic reactions at room temperature, or in concentrated caustic NaOH solution (pH = 14, 82 °C) that simulated a highly accelerated galvanic reaction. In contrast, bismaleimide (BMI) carbon fiber reinforced composites manifested degradation which was aggravated by attempts at toughening the bismaleimide (BMI) matrix. They [208] concluded that the presence of moisture, salts and contact between unprotected metals and BMI carbon fiber composites were necessary but not sufficient conditions for the bismaleimide (BMI) matrix degradation, and attributed its degradation at a significant rate to the generation and build-up of the OH^−^ species generated by galvanic reactions in localized spots. Imide-type carbon fiber composites (like BMI’s and polyimides) have been known [208,347,348,349,350] to degrade as a result of attack by hydroxyl ion species (OH^−^) generated on the surface of the composites which becomes the cathode on galvanic coupling to active metals due to cathodic galvanic reactions in the presence of salts and moisture.

Similar studies [338,339,340,341,342,343] on epoxy based carbon fiber reinforced composites have confirmed that an epoxy matrix is more resistant to degradation under these conditions and requires the presence of both a high pH and a cathodic polarization. The fact that whereas bismaleimide (BMI) matrix degradation can occur in the presence of a high pH alone [208], while epoxy matrix degradation needs both a high pH and a cathodic polarization [341] is indicative of differences in their respective degradation mechanisms. Consequently, degradation of the epoxy matrix have been attributed [340] to the build-up of deleterious concentrations of highly reactive perhydroxyl ions (HO_2_^−^) in occluded regions which can cause damage to both hydrolyzable and non-hydrolyzable polymers. In the hydrolyzable polymers (e.g., esters and imides) which bismaleimide (BMI) belongs to, in addition to the above mechanism, degradation can occur due to nucleophilic attack of hydrolyzable polymer linkages by hydroxyl ions (OH^−^) in alkaline media. With the application of a cathodic polarization, hydroxyl ions (OH^−^) concentrations near bismaleimide (BMI) is bound to increase further and thus account for the much higher degradation observed in bismaleimide (BMI) matrix composites. The resistance of epoxy-matrix to degradation under these conditions can partly explain its use in advanced composites. Since the presence of hyrolyzable species in the polymer matrices account for a greater part of the degradation, the design and development of polymer matrices that are highly resistant to degradation under cathodic polarization will be better served by focusing on systems with little or none of these hydrolyzable links or blending or processing methods that either removes or stabilizes these links in the presence of the deleterious chemical species.

## 10. CFRP Degradation under Anodic Polarization

Under anodic polarization in aqueous electrolytes, which can arise in practice from stray currents, CFRP can be susceptible to degradation as well. Sloan and Talbot [351] have reported that anodic polarization of graphite/epoxy composites at applied current densities as low as 1 μA/cm^2^ are capable of causing rapid and substantial corrosion damage, with the graphite-reinforcing fibers attacked by atomic oxygen that is produced as an intermediate in the oxygen evolution reaction. However unlike metals that degrade under anodic polarization by electrochemical dissolution, CFRP degradation under this condition is due to secondary chemical reactions involving electrochemically evolved species (adsorbed atomic oxygen; an intermediate in the oxygen evolution reaction) which have been linked to oxygen evolution at the (CFRP) anode.

Stafford et al. [352] studied electrochemical stability of graphite fiber–polymer matrix composites as electrolysis electrodes in simulated seawater electrolyte and reported electrochemical and mechanical stability as a cathode but significant interfacial attack directed toward the fiber/resin interface with about 15% reduction in the average fiber diameter as an anode. This interfacial loss was attributed to a combination of chemical and electrochemical graphite oxidation and cavitations due to the evolution of oxygen and chlorine, and almost entirely to oxidation by active chlorine in acidic chloride solutions at moderate current densities, since the current efficiency for chlorine evolution is close to 100% under these conditions. According to this work [352], the increased graphite oxidation is generally observed at pH greater than 4 depending on the temperature and chloride concentration of the electrolyte. Furthermore the initial capacitance increase (during the first few hours of immersion) was reported [352] for carbon fiber–polymer matrix composites under anodic polarization. This initial change of capacitance was attributed to a combination of fiber roughening and surface oxidation. However after the initial period the capacitance tended towards initial values and ultimately decreased. Pittman et al. [353] reported that on continuous application of anodic potentials to high strength PANI-based carbon fibers in 1% KNO_3_ solutions, acidic functional surface groups are produced with the quantity increasing with the extent of the electrochemical oxidation. Bismarck et al. [354] studied the surface properties of PAN-based carbon fibers after anodic oxidation by cyclic voltammetry in different alkaline electrolyte systems, and reported increased fiber surface tension due to increased number of oxygen containing surface groups and roughening of the fiber surfaces as a result of their anodic oxidation. Anodic oxidation of fiber surface is one of the methods used to improve the fiber–matrix adhesion for improved load transfer from the matrix to the fibers [179]. 

Taylor and Humffray [204] studied oxygen reduction on glassy carbon electrodes in solutions of high pH (pH > 10) and reported effects on the mechanism based on the type of prior treatment (cathodic or anodic) given to the glassy carbon electrodes. The major difference in mechanism of oxygen reduction was that oxygen reduction to OH^−^ rather than to peroxide is boosted at all potentials in glassy carbon electrodes given a prior anodic treatment. This report might have important implications on the electrochemical activity and chemical stability of CFRP towards oxygen reduction with respect to the method(s) employed in processing its reinforcing carbon fibers, since the non-production of the reactive peroxides on oxidized carbon fiber surface translates to the presence of less of the species that promote CFRP degradation. 

From the review of the literature above, it is obvious that both the carbon fibers and the polymer matrices of carbon-fiber reinforced polymers are subject to degradation to varying degrees under cathodic polarization. As a consequence there is a need for the development of appropriate methods and procedures to detect, monitor, and predict degradation in fiber reinforced polymer composites since they are often employed in critical applications.

## 11. Monitoring and Mitigating Degradation in Carbon Fiber Reinforced Composites.

Since carbon fiber reinforced composites are used in structural applications similar to metals, and replacing metals in some applications four important questions emerge; how can composite damage be monitored? Can monitoring procedures hitherto applied to metals and alloys be extended/adapted to monitor composite degradation? How can composite degradation be mitigated, especially in applications in which cathodic polarization is feasible in the presence of aqueous media? We will attempt to provide some insights with respect to these important technologically relevant questions based on published literature [343,355] and observations from recent research carried out in our laboratory [278,356].

Bal [232] employed the increase and shifting in the G-band of carbon in the Raman spectra of carbon nanofiber/epoxy composites to confirm stress transfer [357] and reinforcement between epoxy matrix and the carbon nanofibers which might be explored for possible application in the monitoring of degradation in CFRPs.

Taylor et al. [343] studied electrochemical damage in bismaleimide/graphite fiber composites in aerated 0.6 M NaCl and 0.1, 1.0, and 2.0 M NaOH solutions using mainly electrochemical impedance spectroscopy and reported that cathodic polarization of graphite fiber composites produced porous electrode behavior which was attributed to breakdown of the fiber/matrix interface and subsequent moisture intrusion whereas exposure to caustic solutions yielded no porous electrode response leading them to conclude that the reaction intermediates generated during the oxygen reduction reaction, such as peroxide and superoxide radicals, are the main species responsible for degradation and not OH^−^ ions. Their conclusion of fiber/matrix breakdown (Figure 5) was based on the observations of increase in double layer capacitance with time coupled with a decrease in impedance modulus with time, consistent with an increasing electrochemically active area. Absence of fiber surface roughening, the presence of small gaps between fiber and matrix (both observed from scanning electron microscopy), and decreasing slope of the impedance magnitude plot coupled with decreasing phase angle peak are indicative of the “squaring effect” that is characteristic of porous electrodes [358,359,360]. Based on their analysis of impedance spectroscopy data, they [343,355] concluded that the phase angle was the most sensitive indicator of changes (damage) in the composite, and employed “delta phase angle” plots to compare interfacial changes (or material damage) as a function of exposure conditions (exposure time and cathodic polarization) with interesting results. The delta phase angle plot is a frequency by frequency subtraction of the phase angle value at an exposure condition (e.g. an exposure time (t = t_1_) or polarization (η = η_1_)) from a baseline exposure condition (exposure time (t = t_0_) or polarization (η = η_0_)); smaller values of the output tending to zero being indicative of smaller interfacial changes from initial conditions and higher values suggesting more profound changes. A time based “delta phase angle” plot may be calculated or expressed thus [343]:(8)$$\Delta \theta_{\left(\omega_{t=t1}\right)}=\Delta \theta_{\left(\omega_{t=t0}\right)}-\Delta \theta_{\left(\omega_{t=t1}\right)}$$
where *θ* is the phase angle, ω is the frequency and t is time.

Their delta phase angle plots showed that cathodic polarization produces a delta-phase angle plot with characteristic peak between 10 and 100 Hz and a gradually increasing value below 1 Hz referred to as a low frequency “tail”. The peak was attributed to parallel shift of the phase angle response to lower frequencies with increasing damage (increasing time or potential), and the tail to decreased impedance and accompanying phase angles at lower frequencies. On the strength of experimental evidence of the absence of these “tails” in tests in caustic solutions (with abundance of OH^−^) and their emergence in same solutions on addition of H_2_O_2_ without cathodic polarization, these tails were associated to the accumulation of cathodically produced electro-active species which are most probably electrochemically generated peroxide and peroxide intermediates (e.g., superoxide radicals) which were not present in the caustic solutions [343].

The suggested method based on the phase angle evolution can be a way to indicate the changes which occur at the carbon interface, but has no physical background for quantification of the degradation effects. In spite of this reservation, we applied the phase angle method of Taylor [343,355] to our electrochemical impedance data acquired under different test conditions, and observed interesting results which will be the subject of an oncoming communication [361]. However, presented in Figure 6 are the results from electrochemical impedance spectroscopy (EIS) data acquired from CFRP samples immersed in 50 mM NaCl for different time intervals, and at different cathodic polarizations, and treated to obtain the delta phase angles (Δ*θ*) in an attempt to monitor any interfacial degradation in the CFRP due to the various applied cathodic polarizations. It can be observed in Figure 6 that at open circuit potential (OCP) and at 0 mV SCE (measured with reference to saturated calomel electrode), irrespective of the immersion time the delta phase angle (Δ*θ*) was virtually flat at all frequencies which should be indicative of insignificant interfacial degradation. However, as the polarization is increased marked increase is observed in the delta phase angle (Δ*θ*) values at lower frequencies (≤10 Hz). Furthermore, at higher cathodic polarizations (≥750 mV) the increase in the delta phase angle (Δ*θ*) with immersion time is quite obvious which might be indicative of an apparent progressive and cumulative interfacial damage. The apparent high values of the delta phase angle (Δ*θ*) at ≈ −250 mV SCE is attributed to the 2-electron oxygen reduction reaction (Equation (1)) which has been observed [362] to be quite intensive around this cathodic potential. The time independence of the trends of the delta phase angle (Δ*θ*) at this cathodic potential (≈−250 mV SCE) was not expected.

Though we initially had reservations on the exclusive use of phase angle difference from limited number of tests to monitor degradation, as the phase angle is not an independent variable, on the strength of our results and observations, we concede that this approach can be exploited by using enlarged datasets and neural networks to monitor patterns and enhance predictive accuracy in monitoring interfacial degradation of advanced polymer composites. From our experience, damage to the carbon fiber reinforced polymer can be better monitored using electrochemical impedance spectroscopy and making measurements in the absence of polarization (either impressed or by galvanic coupling to metals). Employing this procedure, a constant capacitance on periodic testing might be indicative of the absence of degradation, since degradation (interfacial degradation which can lead to loss of structural integrity as the matrix’s ability to transfer load to the reinforcing carbon fibers is compromised) is likely to result in ingress of solution between the matrix and carbon fibers thus increasing the electrochemically active surface area. However, this might not be valid if in-between measurements species are introduced (on purpose or inadvertently) that may interfere with the electrochemical response of the carbon surface. Fortunately, this occurrence (presence of species that affect the electrochemical response of carbon) may be monitored by comparison of the resistance component (assigned to charge transfer) in the impedance spectra acquired at different time intervals. A constant or fairly constant resistance is indicative of nil or insignificant surface modification while increasing or decreasing resistance can be indicative of surface modification [363]. It should be noted however, that in situations in which the fiber breakage is plausible, increase in resistance might be indicative of a loss of electrical continuity in broken fibers. Hence, electrochemical impedance spectroscopy can be employed to monitor the mechanical damage in carbon fiber reinforced polymer composites as demonstrated in several recent works [364,365,366,367]. 

Figure 7 illustrates the decrease in the electrochemically active surface area of CFRP that can result from reduction of the effective carbon fiber surface area either by surface modification or by a loss of electrical continuity in mechanically damaged fibers. The obvious similar effects of both factors on the electrochemically active surface area explains why electrochemical impedance spectroscopy can be exploited in both scenarios. In spite of the fact that both scenarios produce the same effect (decrease in electrochemically active surface area) on CFRP samples, we presume that the two scenarios might be discriminated by monitoring the evolution of the capacitive component of the measured impedance without impressed potential. Under this test condition, we postulate that in the case of reduction in electrochemically active area due to adsorption of species to carbon fiber surface or surface modification that produce blockage effects, decrease in capacitance with time is more likely whereas in the case of loss of electrical continuity due to mechanical damage capacitance is expected to vary much less with time.

The challenge in attempting to mitigate carbon fiber reinforced polymer degradation due to galvanic coupling to metals can be appreciated from Figure 2 showing the cathodic branch of the potentiodynamic polarization curve of carbon fiber reinforced polymer vis-a-vis the usual ranges of corrosion potentials for selected metals in aqueous media. It can be observed that for iron, its corrosion potential lies in the charge transfer controlled region of the polarization curve of CFRP. Consequently, on coupling iron to CFRP though the cathodic polarization on the carbon fiber reinforced polymer can be less severe because of lower oxygen reduction current densities. Secondly, the potential range to which CFRP is polarized being close to the potential for the 2-electron oxygen reduction reaction that produces the deleterious peroxide species (Equations (1), (6), and (7)) matrix degradation might be a possibility even at this seemingly benign cathodic polarization [278]. For the extreme case of magnesium coupled to CFRP, it can be observed that the CFRP will be polarized to potentials well beyond the potential ranges in which cathodic activity on it is under oxygen diffusion control and hydrogen reduction can already start. Even in the intermediate cases of galvanic coupling to zinc and aluminum in which the CFRP is polarized to potential ranges that cathodic processes on its surface are under diffusion control, the measured cathodic current densities in these potential ranges are quite significant, and reported to be in the range of ≈ 40 μA cm^−2^ in quiescent near-neutral chloride aqueous chloride media, and estimated to be capable of supporting anodic dissolution rates of about 0.599, 0.436, and 0.132 mm yr^−1^ for zinc, aluminum, and iron respectively, assuming equal cathodic and anodic surface areas [278]. Since the electrochemically active carbon phase in the CFRP is not metallic, diminution of electrochemical active surface area by interaction with organic inhibitors as observed in metals (particularly transition metals), in which their d-shell electrons interact with pi-electrons of the hetero-atoms in heterocyclic organic compounds may not apply to carbon, in spite of its well known adsorbent properties [368,369]. Hence to utilize an adsorbate or species in solution to mitigate cathodic activities on carbon surfaces on CFRP, we postulated [356] that the adsorbate must be able to either increase the resistance to charge transfer processes (electron transport across the interface) resulting in lower cathodic current densities, and/or suppress oxygen transport to the surface resulting in lower limiting diffusion current densities, or interfere with the mechanism or mechanistic step(s) of cathodic processes on carbon fiber surfaces. This is not helped by the fact that the reinforcing carbon fibers being small (≈ 6 μm in our studies) may act as microelectrodes with enhanced diffusion implications with respect to fraction of surface attributable to the carbon fibers. In spite of these challenges, some reduction of the cathodic activity on cathodically polarized carbon-fiber reinforced polymers have been reported using a surfactant [278,356] and also with precipitating inhibitors [278,356]. The inhibitive effects were attributed to surface modification by adsorption to carbon surface in the former, and to the emergence of a barrier to oxygen diffusion to CFRP surface consequent on inhibitor precipitation onto CFRP surface due to the local high pH near the CFRP surface.

## 12. Conclusions

From the review of literature above, it has been demonstrated that the evolution of research efforts at improving the mechanical properties of carbon-fiber reinforced polymers, which is focused on the addition of conductive micro- and nano-fillers, is bound to enhance electrical properties of the composite with serious implications for galvanic corrosion of active metals coupled to CFRP in multi-material combinations. This review of the literature suggests that enhanced conductivity in modified carbon-fiber reinforced polymers occur via two principal mechanism; electron hopping at ultralow nano-filler concentrations with well dispersed nanotubes (when the distance between adjacent well-dispersed nano-fillers becomes ≤ the hopping distance of the charge carriers), and percolation at higher nano-filler concentrations with less dispersed CNTs and CNFs. The scope and scale of the problem of galvanically stimulated degradation of CFRPs is projected to increase significantly with increase in the use of CFRP composites as economic materials in a much wider range of applications in which smaller composite parts are prone to come in contact with metals.

With respect to CFRP degradation, from the literature it seems that anodic polarization of CFRP lead to degradation of the carbon fibers (the reinforcing phase) by adsorbed atomic oxygen; an intermediate in the oxygen evolution reaction. In contrast cathodic polarization ensures composite degradation by attack on the epoxy of the matrix phase most probably due to nucleophilic attack by hydroxyl and perhydroxyl ions, formed as a result of the oxygen reduction reaction on carbon fiber. The functional groups on the surface and/or the ratio of basal to edge sites on the carbon surface can exert influences on electrochemical activity of the carbon fibers which in turn can promote or reduce the production of deleterious highly reactive species that can promote degradation of the polymer matrix of the composites. Focus on the development of polymer matrices blends with less or no hydrolzyable links is likely to lead to the emergence of fiber reinforced polymer composites that are less prone to degradation on galvanic coupling to metals. The combination of extensive delta phase angle (Δ*θ*) data bank and a neural network to recognize and match patterns is suggested as a potential non-destructive method for monitoring CFRP degradation in critical infrastructure.

## Figures and Tables

**Figure 1 materials-12-00651-f001:**
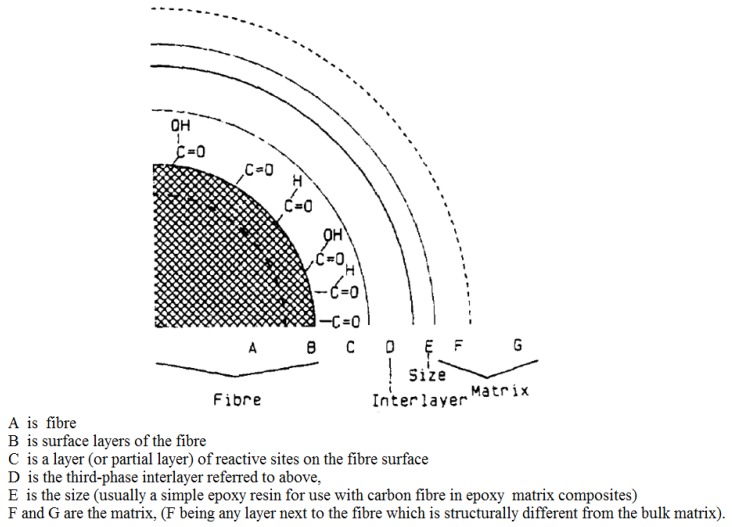
Regions of the carbon fiber-epoxy interface (Reproduced by permission from Elsevier from Ref. [171]).

**Figure 2 materials-12-00651-f002:**
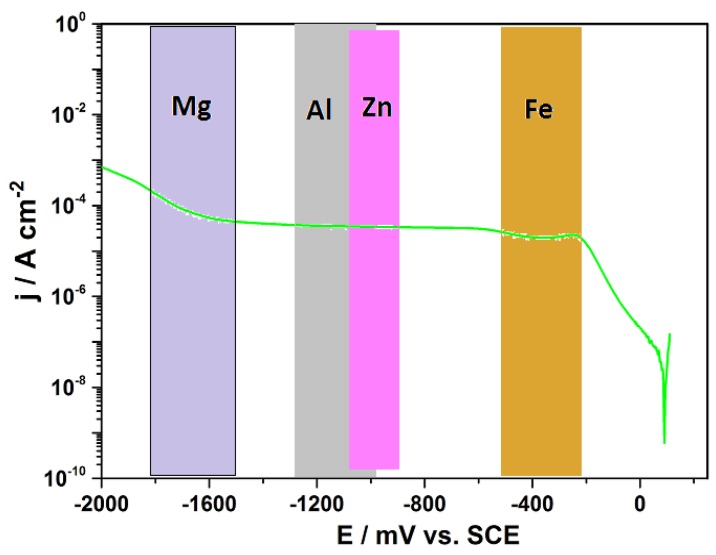
Cathodic potentiodynamic polarization curve of carbon fiber reinforced polymer acquired at a scan rate of 1 mV s^−1^ in quiescent 50 mM NaCl showing the potential ranges each metal can polarize CFRP and the operative cathodic current densities on CFRP in these potential ranges.

**Figure 3 materials-12-00651-f003:**
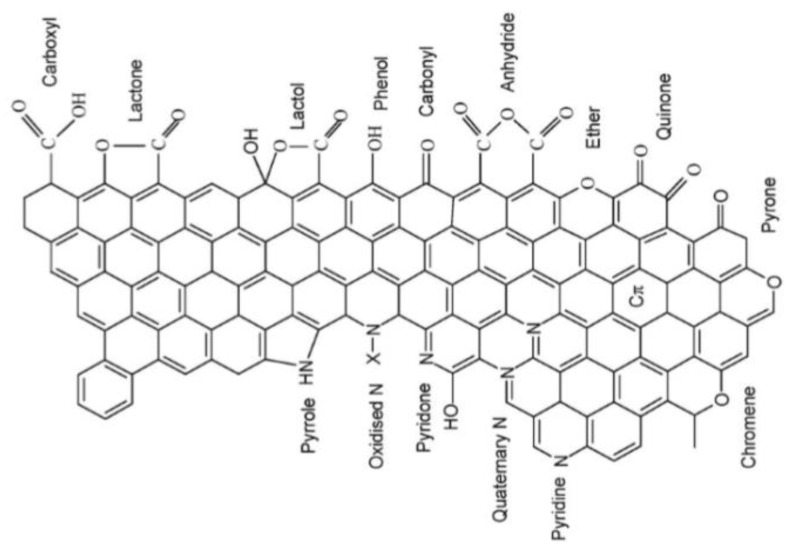
Possible functional groups on carbon surfaces (Reproduced by permission from Elsevier from Ref. [291]).

**Figure 4 materials-12-00651-f004:**
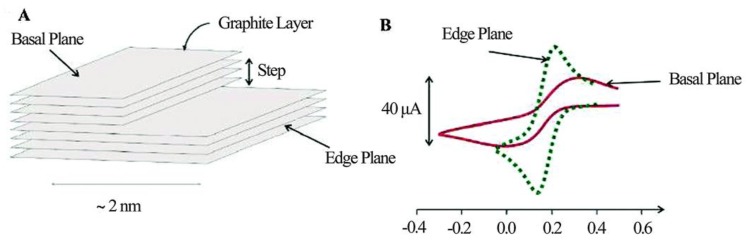
Illustration of edge and basal plane sites on carbon materials and their different electrochemical response (Reproduced by permission from American Chemical Society from Ref. [324]).

**Figure 5 materials-12-00651-f005:**
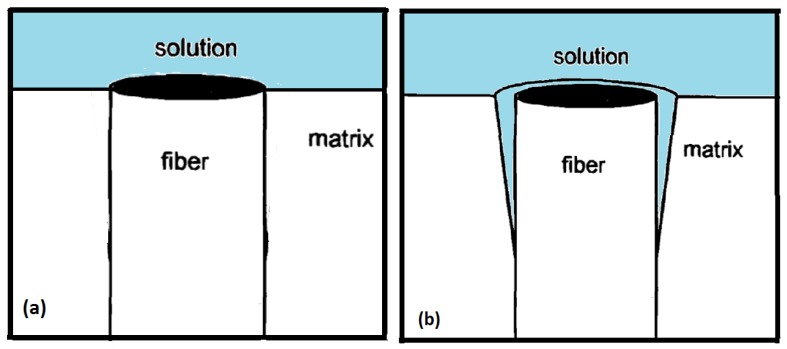
Schematic illustration of proposed model for the origin of the porous electrode effect in cathodically polarized carbon-fiber reinforced polymers (CFRP) samples (**a**) Intact CFRP sample (**b**) Degraded CFRP sample. (Adapted by permission from Electrochemical Society from Ref. [343]).

**Figure 6 materials-12-00651-f006:**
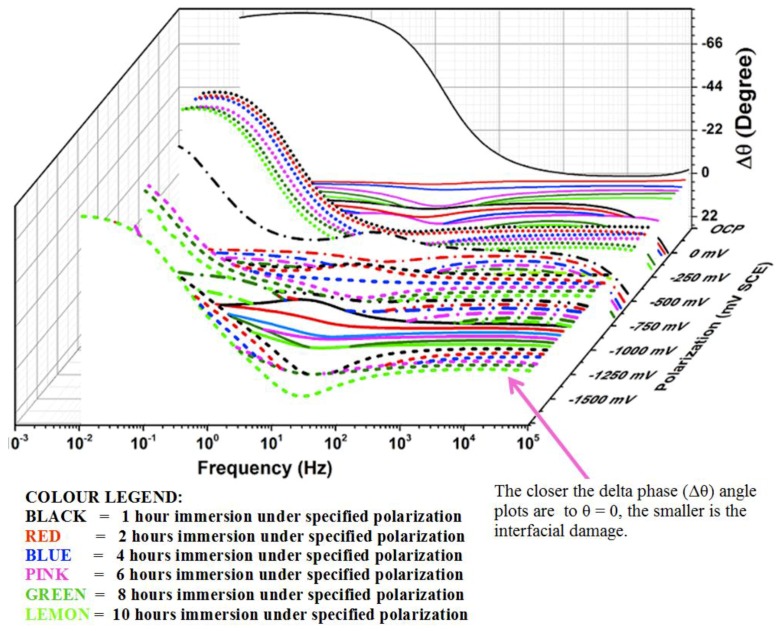
Delta Phase angle plots for CFRP in 50 mM NaCl at different times and cathodic polarizations made with respect to the phase angle after 1 h immersion at open circuit potential (OCP) (first plot in black solid line) to monitor interfacial degradation of the carbon fiber and epoxy interface (after Taylor Refs. [343] and [355]).

**Figure 7 materials-12-00651-f007:**
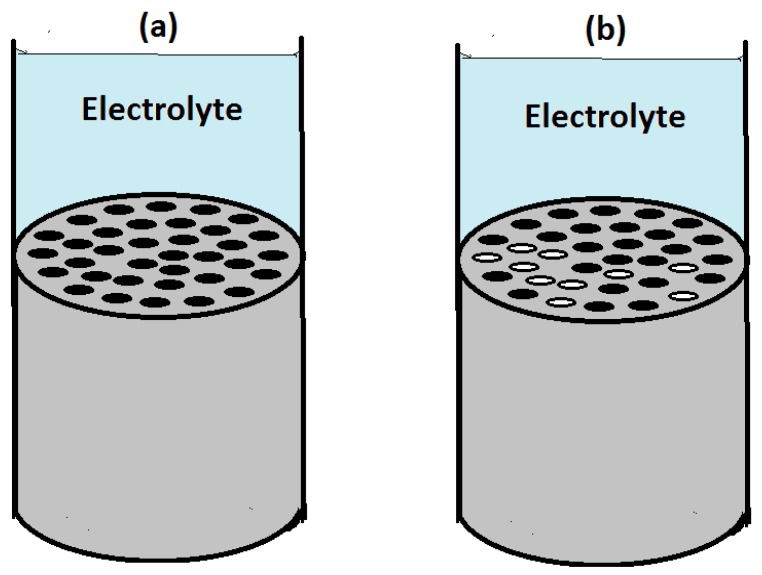
Schematic illustration of the change in the electrochemically active surface area that can result from reduction of the effective carbon fiber surface area either by surface modification or by a loss of electrical continuity in damaged carbon fibers, (**a**) Intact CFRP sample (**b**) CFRP sample with electrochemically active surface area diminished either by surface modification or breakage of carbon fibers (In black are the carbon fibers, white are damaged/surface modified fibers, and in grey is the matrix).

**Table 1 materials-12-00651-t001:** Table of polymeric matrices commonly employed in aerospace sector (Reprinted by permission from Nature/Springer/Palgrave from Ref. [101]). PPS: polyphenylene sulfide; PEEK: polyether ether ketone.

Thermosets				Thermoplastics
Forms Cross-Linked Networks in Polymerization Curing by Heating				No Chemical Change
Epoxies	Phenolics	Polyester	Polyimides	PPS, PEEK
Most popular 80% of total composite usage Moderately high temp. Comparatively expensive	Cheaper Lower viscosity Easy to use High temp usage Difficult to get good quality composites	Cheap Easy to use Popular for general applications at room temp	High temp application 300 °C Difficult to process Brittle	Good damage tolerance Difficult to process as high temp 300-400 °C is required
Low shrinkage (2–3%); No release of volatile during curing	More shrinkage Release of volatile during curing	High shrinkage (7–8%)		
Can be polymerized in several ways giving varieties of structures, morphology and wide range of properties	Inherent stability for thermal oxidation. Good fire and flame retardance Brittle than epoxies	Good chemical resistance Wide range of properties but lower than epoxies. Brittle Low *T_g_*		
Density (g/cm^3^) 1.1–1.4	Density (g/cm^3^) 1.2–1.4	Density (g/cm^3^) 1.1–1.4		Density (g/cm^3^) 1.3–1.4
Tensile modulus 2.7–5.5 GPa	Tensile modulus 2.7–41 GPa	Tensile modulus 1.3–4.1 GPa		Tensile modulus 3.5–4.4 GPa
Tensile strength 40–85 MPa	Tensile strength 35–60 MPa	Tensile strength 40–85 MPa		Tensile strength 100 MPa

**Table 2 materials-12-00651-t002:** Table of reported values of electrical conductivity for modified carbon-epoxy composites (* denotes conductivity values reported in S/m: an and un-an denotes measurements on annealed and un-annealed samples; where a frequency is indicated conductivity values were read from the conductivity spectra at the indicated frequency). CNF: carbon nanofiber; MWCNT: multi-walled carbon nanotube; SWCNT: single-wall carbon nanotubes.

Composite System	Filler Used	Filler wt %	Electrical Conductivity (S/cm)	Reference
CNF-Epoxy matrix	CNF	0.1–1	2 ×10^−6^ to 4 ×10^−3^	Bal (2010) Ref. [232]
CNT-polystyrene	MWCNT	0.5–4	(1 × 10^−10^ to 1 S/cm) (1 × 10^−8^ to 1 S/m)*	Cipriano et al. (2008) Ref. [264]
CNF-polystyrene	CNF	3–15	1 × 10^−10^ to 10^−2^ S/cm ^(an)^ 1 × 10^−10^ to 10^−5^ S/cm ^(un-an)^ (1 × 10^−8^ to near 1 S/m)*	Cipriano et al. (2008) Ref. [264]
CNT-polycarbonate	MWCNT	0.75–3	1 × 10^−14^ to > 10^−2^ S/cm (at 1 Hz)	Pötschke et al. (2004a) Ref. [266]
CNT-polycarbonate	MWCNT	0.875	≈ 10^−3^ S/cm (post annealing at 230 °C)	Alig et al. (2007) Ref. [268]
CNT-polycarbonate	SWCNT MWCNT		1 × 10^−13^ to > 10^−3^ S/cm (at 1 Hz during mixing)	Pötschke et al. (2004b) Ref. [269]
CNT-polycarbonate	MWCNT	0–5	1 × 10^−16^ to ≈10^−2^ S/cm	Pötschke et al. (2003) Ref. [270]
CNT-epoxy CNF-polypropylene CNF-polyethylene	SWCNT CNF CNF	0–40 0–40 0–40	1 ×10^−16^ to ≈ 10^−7^ S/cm 1 × 10^−17^ to ≈ 10^−7^ S/cm 1 × 10^−17^ to ≈ 10^−7^ S/cm	Lozano et al. (2001) Ref. [264]

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
