# Peer review of "Galvanically Stimulated Degradation of Carbon-Fiber Reinforced Polymer Composites: A Critical Review"

_materials, 2019, doi:10.3390/ma12040651_

Reviewer 1 Report

This review treats a scientifically and industrially very important topic. The issue of galvanic corrosion of CFRP in particular in view of novel nanofiller approaches is very timely. In this paper, the basic principles of CFRP are explained first and the use of different filler types is reviewed. The effect of fillers on conductivity is discribed in detail. Mechanisms of galvanically stimulated CFRP degradations are reviewed and eventually the use of the delta phase angle method is discussed.

While parts of the review are very well written and a good literature overview is given, other parts appear more incoherent.

Specifically the role of fillers does not seem clear to me. While it is obvious that CNT or CNF can enhance the conductivity of the composite, I can find no evidence in the review that their addition would actually negatively effect CFRP behaviour with respect to galvanic degradation. It seems to be rather a claim that the authors are making, since also in their own results they do not present any experimental comparison of CFRP with and without fillers.

This is somewhat disappointing after the detailed introduction on CNT/CNF and their role in the composite.

Furthermore I think that the delta phase angle method takes a disproportionally large space. Certainly it is normal that authors of a review will explain their own work in more detail than that of others. But this part does not fit in style and richness of detail with the other sections. I strongly suggest to cut it and also to cut on the diagrams here.

Another source of confusion is the mentioning of 3D printing very prominently in the abstract and in the conclusion, while no galvanic degradation studies of 3D printed material is presented. The only issue with 3D printing seems to be the fact that more CFRP might be around due to the greater ease of production. The topic of 3D printing should therefore be removed at least from the abstract, because it is misleading.

In order to make the review fit for publication I suggest the following:

1) make it very clear that there is no evidence for the consequences of nanofillers on galvanically stimulated degradation of CFRP presented, but only plausibility considerations

2) Section 4: title is misleading. "Possible consequences on galvanically stimulated degradation of CFRP" are not discussed, only percolation and enhanced electrical conductivity. Title should read something like "Efforts at modification of CFRP by fillers and possible consequences for conductivity".

3) Section 4, beginning: clearly distinguish between CNT and CNF. Especially in the intro the two are occasionally mixed up.

4) page 9 second and third paragraph: clay fillers and SiC seem to be off topic, because no electrical conductivity increase is reported. Same for rubber particles in paragraph 2 on page 10. These sections should be cut, because anyways all the discussion is for CNT and CNF later on.

5) page 14, last paragraph: the authors write that Bal found an increase in conductivity at low CNF concentrations, lower than the percolation threshold - but the percolation threshold was taken as 5-20% from carbon black. Certainly carbon black and carbon nanofibers cannot be compared, because CNF are very elongated and the percolation threshold must be much lower than in the case of carbon black. Please provide a reference for the percolation threshold of CNF specifically.

6) page 27-29: reduce diagrams and detail.

7) remove prominent reference to 3D printing from abstract and conclusion, as it is misleading there.

Some detailed comments and edits are found in the attached pdf.

Author Response

REVIEWER 1

RESPONSES TO REVIEWERS' COMMENTS -- Galvanically Stimulated Degradation of CFRP Composites

            Before responding to the comments, I wish to express my immense gratitude to the esteemed reviewers for their time and diligence in going through our manuscript and for the very helpful comments and painstaking suggestions.

REVIEWER 1

Open Review

(x) I would not like to sign my review report 

English language and style

 (x) Moderate English changes required 

 Comments and Suggestions for Authors

This review treats a scientifically and industrially very important topic. The issue of galvanic corrosion of CFRP in particular in view of novel nanofiller approaches is very timely. In this paper, the basic principles of CFRP are explained first and the use of different filler types is reviewed. The effect of fillers on conductivity is described in detail. Mechanisms of galvanically stimulated CFRP degradations are reviewed and eventually the use of the delta phase angle method is discussed. 

REVIEWER 1, COMMENT #1

While parts of the review are very well written and a good literature overview is given, other parts appear more incoherent. 

Specifically the role of fillers does not seem clear to me. While it is obvious that CNT or CNF can enhance the conductivity of the composite, I can find no evidence in the review that their addition would actually negatively effect CFRP behaviour with respect to galvanic degradation. It seems to be rather a claim that the authors are making, since also in their own results they do not present any experimental comparison of CFRP with and without fillers. 

This is somewhat disappointing after the detailed introduction on CNT/CNF and their role in the composite. 

RESPONSE TO REVIEWER 1, COMMENT #1

The role of nanofillers used to modify CFRPs on galvanic corrosion of coupled metal/alloys have now been added. There are limited number of reports on this and this work can help draw attention to this

REVIEWER 1, COMMENT #2

Furthermore I think that the delta phase angle method takes a disproportionally large space. Certainly it is normal that authors of a review will explain their own work in more detail than that of others. But this part does not fit in style and richness of detail with the other sections. I strongly suggest to cut it and also to cut on the diagrams here.

RESPONSE TO REVIEWER 1, COMMENT #2

In response to this important observation only 1 delta phase angle plot which in needed ti highlight the method  is retained in this revised version of the manuscript.

REVIEWER 1, COMMENT #3

Another source of confusion is the mentioning of 3D printing very prominently in the abstract and in the conclusion, while no galvanic degradation studies of 3D printed material is presented. The only issue with 3D printing seems to be the fact that more CFRP might be around due to the greater ease of production. The topic of 3D printing should therefore be removed at least from the abstract, because it is misleading. 

RESPONSE TO REVIEWER 1, COMMENT #3

3D printing has now been removed from both the abstract and conclusions.

REVIEWER 1, COMMENT #4

In order to make the review fit for publication I suggest the following:

1) make it very clear that there is no evidence for the consequences of nanofillers on galvanically stimulated degradation of CFRP presented, but only plausibility considerations

RESPONSE TO REVIEWER 1, COMMENT #4

Reports from 3 recent articles on consequences of nanofillers on galvanically stimulated corrosion of coupled metals have now been added

REVIEWER 1, COMMENT #5

2) Section 4: title is misleading. "Possible consequences on galvanically stimulated degradation of CFRP" are not discussed, only percolation and enhanced electrical conductivity. Title should read something like "Efforts at modification of CFRP by fillers and possible consequences for conductivity". 

RESPONSE TO REVIEWER 1, COMMENT #5

Section 4 title has now been changed from "Efforts at Modification of CFRPs and Possible Consequences on Galvanically Stimulated Degradation of Carbon Fibre Reinforced Polymer Composites"  to  "Efforts at Modification of CFRPs with Nanofillers and Possible Consequences  for Conductivity and  Galvanic Corrosion of Coupled Metals."

In line with the change in the title the few papers available at the moment on the effect of nanofiller modification of CFRPs on galvanic corrosion of coupled metals have been critically reviewed and added to this section. The added text reads thus:

"              The effect of the introduction of CNTs and CNFs into CFRP as nano-fillers on the galvanic corrosion of technologically relevant metals/alloys coupled to CFRP is an area that require detailed study.  However, there are just  few reports in literature on galvanic corrosion of metals coupled to nano-filler modified fibre reinforced polymers  [275-277]  Ireland et al., [275] investigated galvanic corrosion between aluminium 7075 and glass fibre reinforced polymer (GRFP)  composites modified with carbon nanotubes and reported  statistically significant increase (approximate doubling) of corrosion rate and mass loss rate on  coupling MWCNT/GFRP samples with aluminium 7075 compared to baseline GFRP samples. It is important to note that in the MWCNT/GFRP - AA7075 galvanic couple studied by Ireland et al., [275], the reinforcing fibre (glass fibre) in the GFRP is not electrically conductive in contrast to CFRP in which the reinforcing fibre (carbon fibre) is electrically conductive. Baltzis et al., [276] investigated the performance of austenitic stainless steel 304 (SS304) adhesively bonded to neat epoxy, CNT-modified epoxy and carbon fibre (CF)- reinforced epoxy (CFRRP), and reported that though the incorporation of CNTs increased the galvanic effects, it also retarded uniform corrosion and localised corrosion as the modified adhesives prevented the electrolyte from reaching the substrate. Arronche et al., [277] studied galvanic corrosion between AISI 1018 carbon steel coupled to CFRPs modified with multi-walled carbon nanotubes and reported  that the addition of MWCNTs do not have a statistically significant effect  on the corrosion and mass loss rates compared to unmodified CFRP. Comparing their result [277] which is at variance with other works [275,276] that reported "CNT- induced" increase in galvanic corrosion of metals coupled to fibre reinforced polymer composites, they attributed their non-observance of an increase in metallic (AISI 1018 carbon steel) corrosion rates on galvanic coupling to MWCNTs  modified CFRP  to the already conductive nature of carbon fibre reinforcement with respect to the CNT fillers. This attribution in our opinion is most probably incorrect, because addition of CNTs above its percolation threshold in the polymer matrix is bound to increase the conductivity of the matrix phase of the composite so that the entire CFRP composite surface becomes conductive. Ideally, such increase in the conductive area of CFRP surface on addition of CNTs is expected to lead to an increase in the cathodic area of  MWCNTs  modified CFRP - AISI 1018 carbon steel galvanic couple, which should ordinarily support increased anodic dissolution of the coupled carbon steel.  However, from work carried out in our laboratory [278] which studied galvanic corrosion and inhibition in Al-CFRP, Cu-CFRP, Zn-CFRP, and Fe-CFRP galvanic couples and Fig. 2 of this work, it is obvious that unlike CFRP coupled to Al, Zn, and Mg, at the cathodic potentials CFRP is polarized (around -400 mVSCE) cathodic reactions on CFRP surface might not yet be under diffusion control. Hence increase in the conductive surface area of the CFRP composite by modification with the MWCNTs  might be less likely to translate to enough increase in cathodic activity that can support a statistically significant increase in anodic dissolution of Fe or carbon steel. This can  explain the contrasting trend observed by Arronche et al., [277] for galvanic corrosion between AISI 1018 carbon steel coupled to CFRPs modified with multi-walled carbon nanotubes, as all reports of "CNT-induced" galvanic corrosion of metals coupled to CNT-modified CFRPs involved metals/alloys that are able to polarize CFRP to potentials at which cathodic processes on CFRP surface are under diffusion control. "

REVIEWER 1, COMMENT #6

3) Section 4, beginning: clearly distinguish between CNT and CNF. Especially in the intro the two are occasionally mixed up. 

RESPONSE TO REVIEWER 1, COMMENT #6

This has now been done.

REVIEWER 1, COMMENT #7

4) page 9 second and third paragraph: clay fillers and SiC seem to be off topic, because no electrical conductivity increase is reported. Same for rubber particles in paragraph 2 on page 10. These sections should be cut, because anyways all the discussion is for CNT and CNF later on. 

RESPONSE TO REVIEWER 1, COMMENT #7

These have now been deleted.

REVIEWER 1, COMMENT #8

5) page 14, last paragraph: the authors write that Bal found an increase in conductivity at low CNF concentrations, lower than the percolation threshold - but the percolation threshold was taken as 5-20% from carbon black. Certainly carbon black and carbon nanofibers cannot be compared, because CNF are very elongated and the percolation threshold must be much lower than in the case of carbon black. Please provide a reference for the percolation threshold of CNF specifically. 

RESPONSE TO REVIEWER 1, COMMENT #8

Percolation threshold of CNF  is affected by many variables like polymer matrix, processing method/procedure so it is difficult to even compare figures. however many of the reported values are blow 1 wt %. Comparison of the values for carbon black and carbon nanofibers was just to sho how much less material is needed to modify non-conductive polymer matrix to become conductive by using nanofillers. 

Percolation threshold of CNF and CNT filled polymer composites can be influenced by a variety of factors; the major factors being the matrix material and the processing technique [Hermant, M. C. (2009). Manipulating the percolation threshold of carbon nanotubes in polymeric composites PhD Thesis, Technische Universiteit Eindhoven, Netherlands. DOI: 10.6100/IR642602].  Hence a reasonable comparison of percolation thresholds for nano-fillers in different polymer matrices and even similar polymer matrix is difficult. Reported percolation thresholds for these nano-fillers (CNTs and CNFs) in polymer composites are predominantly < 1 wt% [Hermant, M. C. (2009). Manipulating the percolation threshold of carbon nanotubes in polymeric composites PhD Thesis, Technische Universiteit Eindhoven, Netherlands. DOI: 10.6100/IR642602// Sandler J K W, Kirk J E, Kinloch I A, Shaffer M S P and Windle A H 2003 Ultra-low electrical percolation threshold in carbon-nanotube-epoxy composites Polymer 44 5893// Guadagno, L., Raimondo, M., Vittoria, V., Vertuccio, L., Lafdi, K., De Vivo, B., Lamberti, P., Spinelli, G. and Tucci, V., 2013. The role of carbon nanofiber defects on the electrical and mechanical properties of CNF-based resins. Nanotechnology, 24(30), p.305704.//Nayak, L., Chaki, T.K. and Khastgir, D., 2014. Electrical percolation behavior and electromagnetic shielding effectiveness of polyimide nanocomposites filled with carbon nanofibers. Journal of Applied Polymer Science, 131(24).// Nayak, L., Chaki, T.K. and Khastgir, D., 2015. Super heat-resistant conductive nanocomposites based on polysulfone–carbon nanofillers. Polymer-Plastics Technology and Engineering, 54(3), pp.315-323.// Kim, Y.J., Shin, T.S., Do Choi, H., Kwon, J.H., Chung, Y.C. and Yoon, H.G., 2005. Electrical conductivity of chemically modified multiwalled carbon nanotube/epoxy composites. Carbon, 43(1), pp.23-30.].

However, values of percolation thresholds for nano-fillers in polymer composites > 1 wt% have been  reported [Nayak, L., Chaki, T.K. and Khastgir, D., 2015. Super heat-resistant conductive nanocomposites based on polysulfone–carbon nanofillers. Polymer-Plastics Technology and Engineering, 54(3), pp.315-323.].

REVIEWER 1, COMMENT #9

6) page 27-29: reduce diagrams and detail. 

7) remove prominent reference to 3D printing from abstract and conclusion, as it is misleading there. 

RESPONSE TO REVIEWER 1, COMMENT #9

The diagrams have been reduced both in number and size

reference to 3D printing in abstract and conclusion have now been removed

REVIEWER 1, COMMENT #10

Some detailed comments and edits are found in the attached pdf. 

RESPONSE TO REVIEWER1, COMMENT #10

The detailed comments and edits in the attached pdf are highly appreciated.

Corrections and edits have been  made to the manuscript

Reviewer 2 Report

Reconsider after major revision (control missing in some experiments)

I recommend that the authors thoroughly proof-read the work prior to re-submission, paying particular attention to the structure of the review; removing excess/irrelevant information and focusing the work based on the title.

The review should be a collection of peer-reviewed work. Unless I'm mistaken, the work presented towards the end of the review (following on from Taylor et al.) has not been published/peer-reviewed, Including this in a review is potentially miss-leading.  

General comments:

-Mixing of English and American spelling. (Fibres – English; optimization – American)

-et al in italics

-Once you have defined an abbreviation, use it.

-Structure is poor. Why is there a literature review of nano-modified composites? Why introduce CNTs/CNFs then report work on SiC and rubber particles?

Doesn’t feel cohesive. Seems to be a mis-match between the composite, interfacial part and the part about galvanic corrosion.
Generally, the English is poor and could be far succinct with greater detail. References are excessive where it is unnecessary ( i.e 15 references for orthopaedic applications of CFRP), and lacking where needed (i.e. when introducing the issues of galvanic corrosion.)

The introduction does not satisfy the reader with the negative effects of galvanic corrosion. Granted, the combination of materials would lead to this, but more could be done in this section to emphasise the importance of this review. 

Line 18: ‘…..electrically conductive nature’

22: Sentence: ‘In addition the implications…’ doesn’t make sense

30 - metals; ; multi-material – delete a semi colon

41: CFRP is not used for lightning strike protection. It is used to reduce structural weight. It has in fact, poor electrical conductivity. Ref. alludes to modified CFRP

48: ‘…that carbon fibre’

49: ‘…have the potential..’

51: ‘compounds

51 – CFRP once defined just use the abbreviation (this is an issue throughout the script)

52: Sentence doesn’t make sense.

54 – latin in italics – i.e., via, in-situ etc (this comment refers to all future instances of this)

55: ‘composites

62- ‘parts is capable’- ARE capable

67: ‘CO2

72: ‘In most of its technological applications, carbon fibre reinforced polymers (CFRPs) are seldom used alone, but as a component of a hybrid structure.’

73 – 80: Far too verbose.

85: They are not necessarily embedded. Sometimes attached. ‘Consolidated’ would be a better word.

90- needs rewording- not coherent

96-100: Poor English: ‘Since these processes occur on the surface of CFRP and can have deleterious effects on the CFRP itself, the cathodic processes occuring on CFRP surface under cathodic potentials similar to that it wll be subjected to on galvanic coupling  with more active metals ,the degradative processes on CFRP, and possibilities for monitoring it are  critically reviewed.’

The degrading cathodic process between CFRP and active metals and how they can be monitored shall be reviewed.

98 – spelling issue – will not wll

107: Remove ‘While…’.

107: Coatings are normally applied to fibres

111: Consistency: use CFRP, not the full name, 

111: ‘keep them fixed..’

114: Not sure what ‘optimize important performance indicators like ductility’ means. You can impart ductility through the fibres.

115: Delete sentence: ‘In order to achieve these functions it is 116 necessary that the matrix has a lower modulus and greater elongation than those of fibres, so that 117 fibres are made to carry maximum load’. It is not correct.

123 remove punctuation before [99,102]

128: εmax? Should this be in parentheses?

129: ‘composite failure takes place..’

148 – polymeric no need for capital P

164: I would remove the subjectivity; delete: and in our opinion capable of displacing epoxy as a matrix of choice in advanced polymer composites for the aerospace industry.

181: This sentence: ‘In a composite material, the interface or interphase is so important that it is generally regarded as the third constituent of a composite material’ is a repeat on line 176.

184: Postulated it as what?

Fig. 1: On diagram ‘E’ is the sizing

201: Comma after ‘interface’

204 – interlaminar one word- be consistent! Some places have used hyphen and others two separate words.

207 – polymer not polymers

208: Comma after ‘specimens’

209: What is ideal? Fragmentation leading to delamination can lead to ductility which can be ideal.

213: Please double check that the interlaminar shear strength and the fibre pull-out is related to each other.

221: Remove bold formatting on text.

228 – don’t understand – a surface more x?

225-231: Too verbose.

242: Change to ‘…have been pre-treated in an oxygen environment’

237-244: Don’t understand.

245-253: Unnecessary and should be deleted.

257-262: Don’t understand.

265- CNT/CNF – should be other way round- consistent with text (this is issue in other places through work). Then once defined use this through text do not alternate.

268-275: CNFs and CNTs become mixed up

310: Move reference to end of sentence.

318- et al requires . after al

318 –‘of effect’- repeated remove

320 amino functionalisation of DWCNTs + MWCNTs

323- already defined DWCNTs – this section should be checked for other errors of this

336 ‘of a of micro-particle’ – delete ‘a of’

365: ‘referred to as a if a so-called percolating network’ ?

407 – ‘ Transmission electron microscopy (TEM)- don’t think transmission requires capital?

438 MWCNTI?

490: ‘metals and alloys’

495 poor quality figure 2?

501 Oxygen doesn’t requie capital unless all words of the title have capital

537 – titles with capitals or not? Decide and stick with it- issue throughout script

600 – ORR not defined till after first appearance of acronym

630: Sentence beginning with ‘The ability of…’ doesn’t make sense.

648- don’t understand

649: Consistency: Abbreviated FTIR, but not SEM.

680: ‘degradation is ’ instead of ‘degrade it is’.

702-707 – Sentence is too long.

728: ‘indicating’ instead of ‘indicates’

760 – now referring to graphite composites- disjoints the paper as previously referred to as carbon- another point of American vs English language issues noticed in paper

774 not sure this sentence makes sense

788: Capitalise ‘Can’

‘We’ is used at various point in the text as this is too informal- also as a review paper it should be compiling research already published thus no ‘we’

805: Sentence beginning with ‘Absence..’ doesn’t make sense.

839: ‘ant’?

830+: A review should be a collection of published (peer-reviewed) work.

Caption for Figure 9: (‘In black’ as opposed to ‘in back’).

956: Replace ‘Figure 1’ with ‘Figure 2’.

956 specie should be species

 Author Response

REVIEWER 2

RESPONSES TO REVIEWERS' COMMENTS -- Galvanically Stimulated Degradation of CFRP Composites

            Before responding to the comments, I wish to express my immense gratitude to the esteemed reviewers for their time and diligence in going through our manuscript and for the very helpful comments and painstaking suggestions.

RVW 2:

Open Review

(x) I would not like to sign my review report 

English language and style

(x) Extensive editing of English language and style required 

REVIEWER 2

Comments and Suggestions for Authors

REVIEWER 2, COMMENT #1

Reconsider after major revision (control missing in some experiments)

RESPONSE TO REVIEWER 2, COMMENT #1

The figures without some controls presented (Figs 7 and 8) have now been withdrawn.

REVIEWER 2, COMMENT #2

I recommend that the authors thoroughly proof-read the work prior to re-submission, paying particular attention to the structure of the review; removing excess/irrelevant information and focusing the work based on the title.

RESPONSE TO REVIEWER 2, COMMENT #2

Materials not very relevant to the title have now been removed.

However, new material on effect of nanofillers in CFRP on galvanic corrosion of coupled metals from the only 3 articles in the literature and very vital to this review have now been added.

REVIEWER 2, COMMENT #3
The review should be a collection of peer-reviewed work. Unless I'm mistaken, the work presented towards the end of the review (following on from Taylor et al.) has not been published/peer-reviewed, Including this in a review is potentially miss-leading. 

RESPONSE TO REVIEWER 2, COMMENT #3

We agree that a review should be a collection of published (peer-reviewed) work. 

All the work presented towards the end of the review (following on from Taylor et al.) have been removed except Figure 6. as it is necessary to highlight the use of delta phase angles to monitor CFRP degradation. It  has been emphasized that this presentation is based on data treatment procedure that had undergone peer-review.

An alternative could be to obtain copyright to reproduce the plots of Taylor et al., but  the plots they presented were few.

In agreement with this position Figures 7 and 8 and the discussions thereof have been removed from this manuscript as they are extracts from an up-coming publication. However, Figure 6 which is strictly speaking the only "yet to be peer-reviewed" material is retained as it is necessary to highlight the use of delta phase angles to monitor CFRP degradation. It  has been emphasized that this presentation is based on data treatment procedure that had undergone peer-review.

REVIEWER 2, COMMENT #4

General comments:

-Mixing of English and American spelling. (Fibres – English; optimization – American)

-et al in italics

-Once you have defined an abbreviation, use it.

RESPONSE TO REVIEWER 2, COMMENT #4

These corrections have now been effected in the manuscript

REVIEWER 2, COMMENT #5

-Structure is poor. Why is there a literature review of nano-modified composites? Why introduce CNTs/CNFs then report work on SiC and rubber particles?

RESPONSE TO REVIEWER 2, COMMENT #5

Thanks for pointing out this error

Attempts have now been made to enhance the structure of the manuscript .

Reports of work on SiC and rubber particles have now been removed.

Works on effect of nanofillers used to modify CFRP on galvanic corrosion of coupled metals have now been included.

REVIEWER 2, COMMENT #6

Doesn’t feel cohesive. Seems to be a mis-match between the composite, interfacial part and the part about galvanic corrosion.

RESPONSE TO REVIEWER 2, COMMENT #6

The  works has now enhanced with new material on the galvanic corrosion and  effect of nanofillers used to modify CFRP on galvanic corrosion of coupled metals.

The few papers available at the moment on the effect of nanofiller modification of CFRPs on galvanic corrosion of coupled metals have been critically reviewed and added to this section. The added text reads thus:

"              The effect of the introduction of CNTs and CNFs into CFRP as nano-fillers on the galvanic corrosion of technologically relevant metals/alloys coupled to CFRP is an area that require detailed study.  However, there are just  few reports in literature on galvanic corrosion of metals coupled to nano-filler modified fibre reinforced polymers  [275-277]  Ireland et al., [275] investigated galvanic corrosion between aluminium 7075 and glass fibre reinforced polymer (GRFP)  composites modified with carbon nanotubes and reported  statistically significant increase (approximate doubling) of corrosion rate and mass loss rate on  coupling MWCNT/GFRP samples with aluminium 7075 compared to baseline GFRP samples. It is important to note that in the MWCNT/GFRP - AA7075 galvanic couple studied by Ireland et al., [275], the reinforcing fibre (glass fibre) in the GFRP is not electrically conductive in contrast to CFRP in which the reinforcing fibre (carbon fibre) is electrically conductive. Baltzis et al., [276] investigated the performance of austenitic stainless steel 304 (SS304) adhesively bonded to neat epoxy, CNT-modified epoxy and carbon fibre (CF)- reinforced epoxy (CFRRP), and reported that though the incorporation of CNTs increased the galvanic effects, it also retarded uniform corrosion and localised corrosion as the modified adhesives prevented the electrolyte from reaching the substrate. Arronche et al., [277] studied galvanic corrosion between AISI 1018 carbon steel coupled to CFRPs modified with multi-walled carbon nanotubes and reported  that the addition of MWCNTs do not have a statistically significant effect  on the corrosion and mass loss rates compared to unmodified CFRP. Comparing their result [277] which is at variance with other works [275,276] that reported "CNT- induced" increase in galvanic corrosion of metals coupled to fibre reinforced polymer composites, they attributed their non-observance of an increase in metallic (AISI 1018 carbon steel) corrosion rates on galvanic coupling to MWCNTs  modified CFRP  to the already conductive nature of carbon fibre reinforcement with respect to the CNT fillers. This attribution in our opinion is most probably incorrect, because addition of CNTs above its percolation threshold in the polymer matrix is bound to increase the conductivity of the matrix phase of the composite so that the entire CFRP composite surface becomes conductive. Ideally, such increase in the conductive area of CFRP surface on addition of CNTs is expected to lead to an increase in the cathodic area of  MWCNTs  modified CFRP - AISI 1018 carbon steel galvanic couple, which should ordinarily support increased anodic dissolution of the coupled carbon steel.  However, from work carried out in our laboratory [278] which studied galvanic corrosion and inhibition in Al-CFRP, Cu-CFRP, Zn-CFRP, and Fe-CFRP galvanic couples and Fig. 2 of this work, it is obvious that unlike CFRP coupled to Al, Zn, and Mg, at the cathodic potentials CFRP is polarized (around -400 mVSCE) cathodic reactions on CFRP surface might not yet be under diffusion control. Hence increase in the conductive surface area of the CFRP composite by modification with the MWCNTs  might be less likely to translate to enough increase in cathodic activity that can support a statistically significant increase in anodic dissolution of Fe or carbon steel. This can  explain the contrasting trend observed by Arronche et al., [277] for galvanic corrosion between AISI 1018 carbon steel coupled to CFRPs modified with multi-walled carbon nanotubes, as all reports of "CNT-induced" galvanic corrosion of metals coupled to CNT-modified CFRPs involved metals/alloys that are able to polarize CFRP to potentials at which cathodic processes on CFRP surface are under diffusion control. "

REVIEWER 2, COMMENT #7
Generally, the English is poor and could be far succinct with greater detail. References are excessive where it is unnecessary ( i.e 15 references for orthopaedic applications of CFRP), and lacking where needed (i.e. when introducing the issues of galvanic corrosion.)

The introduction does not satisfy the reader with the negative effects of galvanic corrosion. Granted, the combination of materials would lead to this, but more could be done in this section to emphasise the importance of this review. 

RESPONSE TO REVIEWER 2, COMMENT #7

Efforts have been made to improve the clarity of the manuscript.

Issues with  introducing the galvanic corrosion part effectively had to do with the dearth of relevant published work which this review seeks to draw attention.

Spurred by your esteemed criticism , 3 important and recent works on the effect of nanofillers on galvanic corrosion of metals coupled to  nanofiller modified  CFRP has now been reviewed and added.

REVIEWER 2, COMMENT #8

Line 18: ‘…..electrically conductive nature’

RESPONSE TO REVIEWER 2, COMMENT #8

"electrically"  has now been inserted

REVIEWER 2, COMMENT #9

22: Sentence: ‘In addition the implications…’ doesn’t make sense

RESPONSE TO REVIEWER 2, COMMENT #9

Sentence has now been re-phrased and  "In addition the implications… "  deleted.

REVIEWER 2, COMMENT #10

30 - metals; ; multi-material – delete a semi colon

RESPONSE TO REVIEWER 2, COMMENT #10

The extra semi-colon has now been deleted.

REVIEWER 2, COMMENT #11

41: CFRP is not used for lightning strike protection. It is used to reduce structural weight. It has in fact, poor electrical conductivity. Ref. alludes to modified CFRP

RESPONSE TO REVIEWER 2, COMMENT #11

 Sentence has now been re-phrased by inserting  "....on modification to enhance electrical conductivity"

REVIEWER 2, COMMENT #12

48: ‘…that carbon fibre’

RESPONSE TO REVIEWER 2, COMMENT #12

"the carbon fibre" has now been re-phrased to read  " that carbon fibre"

REVIEWER 2, COMMENT #13

49: ‘…have the potential..’

RESPONSE TO REVIEWER 2, COMMENT #13

"....have a  potential" has now been corrected to read "...have the potential "

REVIEWER 2, COMMENT #14

51: ‘compounds

RESPONSE TO REVIEWER 2, COMMENT #14

"compound" has now been corrected to read "compounds"

REVIEWER 2, COMMENT #15

51 – CFRP once defined just use the abbreviation (this is an issue throughout the script)

RESPONSE TO REVIEWER 2, COMMENT #15

Use of CFRP after defining  the abbreviation  has now been effected in the entire manuscript

REVIEWER 2, COMMENT #16

52: Sentence doesn’t make sense.

RESPONSE TO REVIEWER 2, COMMENT #16

"A the carbon–polyvinylchloride (C–PVC) composite electrode has been reported [54]" to rad thus "Carbon–polyvinylchloride (C–PVC) composite electrode has been reported [54]"

REVIEWER 2, COMMENT #17

54 – latin in italics – i.e., via, in-situ etc (this comment refers to all future instances of this)

RESPONSE TO REVIEWER 2, COMMENT #17

Correction has been effected accordingly throughout the manuscript

REVIEWER 2, COMMENT #18

55: ‘composites

RESPONSE TO REVIEWER 2, COMMENT #18

"composite" has been  corrected to read  "composites"

REVIEWER 2, COMMENT #19

62- ‘parts is capable’- ARE capable

RESPONSE TO REVIEWER 2, COMMENT #19

"..... is capable...." now corrected to read thus: ".....are capable......."

REVIEWER 2, COMMENT #20

67: ‘CO2

RESPONSE TO REVIEWER 2, COMMENT #20

"CO2" has now been corrected to read; " CO2"

REVIEWER 2, COMMENT #21

72: ‘In most of its technological applications, carbon fibre reinforced polymers (CFRPs) are seldom used alone, but as a component of a hybrid structure.’

RESPONSE TO REVIEWER 2, COMMENT #21

Sentence has now been re-phrased to read thus:  " In most of its technological applications CFRPs are  used  as  component(s) of hybrid structures."

REVIEWER 2, COMMENT #22

73 – 80: Far too verbose.

RESPONSE TO REVIEWER 2, COMMENT #22

Section has been re-phrased by deletion of the phrase " composed of carbon-fibre reinforced polymers (CFRPs)"

REVIEWER 2, COMMENT #23

85: They are not necessarily embedded. Sometimes attached. ‘Consolidated’ would be a better word.

RESPONSE TO REVIEWER 2, COMMENT #23

"embedded" has been replaced with  "consolidated"

REVIEWER 2, COMMENT #24

90- needs rewording- not coherent

RESPONSE TO REVIEWER 2, COMMENT #24

Sentence has now been re-phrased and reads thus: "Irrespective of the application or the process by which galvanic coupling of CFRP with metals is established, the carbon fibres in CFRPs being conductive are able to support electrochemical charge transfer processes leading to anodic dissolution of the metal, and possible sabotage of  structural integrity."

REVIEWER 2, COMMENT #25

96-100: Poor English: ‘Since these processes occur on the surface of CFRP and can have deleterious effects on the CFRP itself, the cathodic processes occuring on CFRP surface under cathodic potentials similar to that it wll be subjected to on galvanic coupling  with more active metals ,the degradative processes on CFRP, and possibilities for monitoring it are  critically reviewed.’

The degrading cathodic process between CFRP and active metals and how they can be monitored shall be reviewed.

RESPONSE TO REVIEWER 2, COMMENT #25

Sentence has now been re-phrased and reads thus: " These processes occur on CFRP surface and can have degradative effects on the CFRP itself, and metals galvanically coupled to it,  these cathodic processes their degradative effects, and possibilities for monitoring them shall be reviewed."

REVIEWER 2, COMMENT #26

98 – spelling issue – will not wll

RESPONSE TO REVIEWER 2, COMMENT #26

"wll" has now been replaced with "will"

REVIEWER 2, COMMENT #27

107: Remove ‘While…’.

RESPONSE TO REVIEWER 2, COMMENT #27

"‘While" has now been deleted.

REVIEWER 2, COMMENT #28

107: Coatings are normally applied to fibres

RESPONSE TO REVIEWER 2, COMMENT #28

Yes, but they do have implications on electrochemical response among others.

REVIEWER 2, COMMENT #29

111: Consistency: use CFRP, not the full name, 

RESPONSE TO REVIEWER 2, COMMENT #29

Consistency in  use of CFRP, not the full name,  after definition has now been implemented all through the manuscript.

REVIEWER 2, COMMENT #30

111: ‘keep them fixed..’

RESPONSE TO REVIEWER 2, COMMENT #30

"Keeping them fixed" now changed to read thus "keep them fixed"

REVIEWER 2, COMMENT #31

114: Not sure what ‘optimize important performance indicators like ductility’ means. You can impart ductility through the fibres.

RESPONSE TO REVIEWER 2, COMMENT #31

Yes ductility can be tuned via the fibres , but other factors are involved and arV not totaly indepedent of each other hence the need to "optimize"

REVIEWER 2, COMMENT #32

115: Delete sentence: ‘In order to achieve these functions it is 116 necessary that the matrix has a lower modulus and greater elongation than those of fibres, so that 117 fibres are made to carry maximum load’. It is not correct.

RESPONSE TO REVIEWER 2, COMMENT #32

Sentence has now been deleted.

REVIEWER 2, COMMENT #33

123 remove punctuation before [99,102]

RESPONSE TO REVIEWER 2, COMMENT #33

punctuation before [99,102] has now been removed

REVIEWER 2, COMMENT #34

128: εmax? Should this be in parentheses?

RESPONSE TO REVIEWER 2, COMMENT #34

εmax  is no enclosed in brackets thus "( εmax)"

REVIEWER 2, COMMENT #35

129: ‘composite failure takes place..’

148 – polymeric no need for capital P

RESPONSE TO REVIEWER 2, COMMENT #35

"composite failure takes place.." now   changed to "composite failure occurs.."

polymer now appears with lowercase "p"

REVIEWER 2, COMMENT #36

164: I would remove the subjectivity; delete: and in our opinion capable of displacing epoxy as a matrix of choice in advanced polymer composites for the aerospace industry.

RESPONSE TO REVIEWER 2, COMMENT #36

The phrase " in our opinion capable of displacing epoxy as a matrix of choice in advanced polymer composites for the aerospace industry " has now been deleted.

REVIEWER 2, COMMENT #37

181: This sentence: ‘In a composite material, the interface or interphase is so important that it is generally regarded as the third constituent of a composite material’ is a repeat on line 176.

RESPONSE TO REVIEWER 2, COMMENT #37

Repeated text have now been deleted.

 REVIEWER 2, COMMENT #38

184: Postulated it as what?

RESPONSE TO REVIEWER 2, COMMENT #38

"illustrated"  has now been used in place of "postulated"

REVIEWER 2, COMMENT #39

Fig. 1: On diagram ‘E’ is the sizing

RESPONSE TO REVIEWER 2, COMMENT #39

Legend for the diagram of Figure 1 has now been corrected .

REVIEWER 2, COMMENT #40

201: Comma after ‘interface’

RESPONSE TO REVIEWER 2, COMMENT #40

A comma has now been added after "interface"

REVIEWER 2, COMMENT #41

204 – interlaminar one word- be consistent! Some places have used hyphen and others two separate words.

RESPONSE TO REVIEWER 2, COMMENT #41

Consistency has now been maintained by adding a hypen in the word all through the text

REVIEWER 2, COMMENT #42

207 – polymer not polymers

RESPONSE TO REVIEWER 2, COMMENT #42

Correction has now been effected by removing "s" from "polymers" so it  reads thus "polymer"

REVIEWER 2, COMMENT #43

208: Comma after ‘specimens’

RESPONSE TO REVIEWER 2, COMMENT #43

A Comma has now been inserted after "specimens"

REVIEWER 2, COMMENT #44

209: What is ideal? Fragmentation leading to delamination can lead to ductility which can be ideal.

RESPONSE TO REVIEWER 2, COMMENT #44

For cases in which maximum strength is required failure by fibre pull-out is the ideal. In some other instances maximum strength might not be the design objective in using CFRPs

REVIEWER 2, COMMENT #45

213: Please double check that the interlaminar shear strength and the fibre pull-out is related to each other.

RESPONSE TO REVIEWER 2, COMMENT #45

They are related

REVIEWER 2, COMMENT #46

221: Remove bold formatting on text.

RESPONSE TO REVIEWER 2, COMMENT #46

Bold formatting on text was an error and has now been removed.

REVIEWER 2, COMMENT #47

228 – don’t understand – a surface more x?

RESPONSE TO REVIEWER 2, COMMENT #47

Has now been corrected to read thus: "a surface more suitable....."

REVIEWER 2, COMMENT #48

225-231: Too verbose.

RESPONSE TO REVIEWER 2, COMMENT #48

Sentences have been re-phrased.  2 sentences now broken into 4 sentences reading thus:

" The observed improvements in the fibre-polymer adhesion due to these treatments have been attributed to three mechanisms in the absence of a consensus [173]. The first mechanism is enhanced adsorption of the resin molecules onto surface complexes formed on carbon fibre surfaces as a result of the pre-treatment(s) with acidic complexes which are considered to be more effective [197,198]. The  second mechanism is the emergence of  a surface more suitable for polymer resin adsorption due to removal of surface contaminants from carbon fibre surface due to the pre-treatment. The third mechanism is the mechanical keying effect that enables the polymer resin to permeate pits and channels formed on the surface of carbon fibres that have undergone the oxidative pre-treatment [173]."

REVIEWER 2, COMMENT #49

242: Change to ‘…have been pre-treated in an oxygen environment’

RESPONSE TO REVIEWER 2, COMMENT #49

Correction has now been effected.

REVIEWER 2, COMMENT #50

237-244: Don’t understand.

RESPONSE TO REVIEWER 2, COMMENT #50

In fabricating the carbon fibres they undergo pre-treatments which have implications on the path /mechanism of ORR on the fibre surface . Treatments that lead to ORR producing OH-  as a product/by-product  are  less harmful than those that lead to production of peroxide which promotes more intense composite degradation .

REVIEWER 2, COMMENT #51

245-253: Unnecessary and should be deleted.

RESPONSE TO REVIEWER 2, COMMENT #51

Sir, this section is actually important as it gives information on important factors that come into play in determining  the strength of the fibre matrix interface: mechanical and chemical factors. It is these chemical factors that are directly and predominantly affected on composite degradation. 

REVIEWER 2, COMMENT #52

257-262: Don’t understand.

RESPONSE TO REVIEWER 2, COMMENT #52

An important word " modified " was omitted.

 Sentence have now been changed to read thus: " Recent trends in the evolution of modified carbon fibre based polymer composites appear to be quadro-pronged;...... " instad of like this " Recent trends in the evolution of  carbon fibre based polymer composites appear to be quadro-pronged;......

REVIEWER 2, COMMENT #53

265- CNT/CNF – should be other way round- consistent with text (this is issue in other places through work). Then once defined use this through text do not alternate.

RESPONSE TO REVIEWER 2, COMMENT #53

This has now been effected.

REVIEWER 2, COMMENT #54

268-275: CNFs and CNTs become mixed up

RESPONSE TO REVIEWER 2, COMMENT #54

Mix-up in the abbreviations have been corrected. Both CNTs and CNFs are used to modify CFRPs and collectively called nanofillers or carbon nanofillers. From number of literature reports on each, CNTs seem to be the favourite.

REVIEWER 2, COMMENT #55

310: Move reference to end of sentence.

RESPONSE TO REVIEWER 2, COMMENT #55

Paragraph has now been deleted

REVIEWER 2, COMMENT #56

318- et al requires . after al

RESPONSE TO REVIEWER 2, COMMENT #56

Correction has now been effected.

REVIEWER 2, COMMENT #57

318 –‘of effect’- repeated remove

RESPONSE TO REVIEWER 2, COMMENT #57

Repeated word " of effect " has now been removed

REVIEWER 2, COMMENT #58

320 amino functionalisation of DWCNTs + MWCNTs

RESPONSE TO REVIEWER 2, COMMENT #58

The use of the abbreviations  DWCNTs and MWCNTs have now been employed after their introduction and definition on first appearance in text

REVIEWER 2, COMMENT #59

323- already defined DWCNTs – this section should be checked for other errors of this

RESPONSE TO REVIEWER 2, COMMENT #59

This error has now been corrected and the abbreviations used instead of both abbreviation and full name after its definition.

REVIEWER 2, COMMENT #60

336 ‘of a of micro-particle’ – delete ‘a of’

RESPONSE TO REVIEWER 2, COMMENT #60

This part of the manuscript has now been deleted.

REVIEWER 2, COMMENT #61

365: ‘referred to as a if a so-called percolating network’ ?

RESPONSE TO REVIEWER 2, COMMENT #61

Correction has now been effected by deleting  " if a"  from the phrase " referred to as a if a so-called percolating network "

REVIEWER 2, COMMENT #62

407 – ‘ Transmission electron microscopy (TEM)- don’t think transmission requires capital?

RESPONSE TO REVIEWER 2, COMMENT #62

Correction has now been effected by changing  capital letter "T" in " Transmission electron microscopy (TEM)" to lower case "t" to now read thus:  " transmission electron microscopy (TEM)" 

REVIEWER 2, COMMENT #63

438 MWCNTI?

RESPONSE TO REVIEWER 2, COMMENT #63

Correction has now been effected by removing the letter "I" in " MWCNTI"  so that it now reads thus:  " MWCNT" 

REVIEWER 2, COMMENT #64

490: ‘metals and alloys’

RESPONSE TO REVIEWER 2, COMMENT #64

A space has now been inserted so that  "metalsand alloys" now reads  thus: "metals and alloys"

REVIEWER 2, COMMENT #65

495 poor quality figure 2?

RESPONSE TO REVIEWER 2, COMMENT #65

Quality of Figure 2 has now been enhanced.

REVIEWER 2, COMMENT #66

501 Oxygen doesn’t requie capital unless all words of the title have capital

RESPONSE TO REVIEWER 2, COMMENT #66

Correction has now been effected  with words in the title in  capital letters thus: " Mechanism of Oxygen Reduction on Carbon Electrodes"

REVIEWER 2, COMMENT #67

537 – titles with capitals or not? Decide and stick with it- issue throughout script

RESPONSE TO REVIEWER 2, COMMENT #67

Titles have now been uniformly formatted to appear in capital title case format all through the manuscript.

REVIEWER 2, COMMENT #68

600 – ORR not defined till after first appearance of acronym

RESPONSE TO REVIEWER 2, COMMENT #68

ORR has now been defined at the appropriate place in Line 520 of the earlier version of manuscript by amending the phrase ""In spite of the fact that a full-scale mechanistic analysis of the oxygen reduction reaction ....." to now read thus: "In spite of the fact that a full-scale mechanistic analysis of the oxygen reduction reaction (ORR)......."

Corrections was also made in accordance with this change by use of "ORR" from line 520 downwards

REVIEWER 2, COMMENT #69

630: Sentence beginning with ‘The ability of…’ doesn’t make sense.

RESPONSE TO REVIEWER 2, COMMENT #69

Sentence have now been modified from "The ability of the 630 carbon fibre component of CFRP composites to support cathodic reactions in multi-material assemblies with 631 metals is capable of affecting the structural integrity of such hybrid structures in two major ways;....." to read thus: "The ability of the electrically conductive carbon fibre component of CFRP composites to support cathodic reactions on galvanic coupling to metals in multi-material assemblies is capable of affecting the structural integrity of such hybrid structures in two major ways;........"

REVIEWER 2, COMMENT #70

648- don’t understand

RESPONSE TO REVIEWER 2, COMMENT #70

The phrase  "Sloan [338] carried out a after long-term environmental exposure (one year) to study the degradation of ....." has now been rephrased to read thus: " Sloan [338] carried out  long-term environmental exposure (one year) to study the degradation of ........"

REVIEWER 2, COMMENT #71

649: Consistency: Abbreviated FTIR, but not SEM.

RESPONSE TO REVIEWER 2, COMMENT #71

Both have now been abbreviated so that the earlier statement  "...using scanning electron microscopy and FT-IR analysis, now reads thus: "....using SEM and FT-IR analysis,"

REVIEWER 2, COMMENT #72

680: ‘degradation is ’ instead of ‘degrade it is’.

RESPONSE TO REVIEWER 2, COMMENT #72

"degrade it is"  has now been corrected to: " degradation is "

REVIEWER 2, COMMENT #73

702-707 – Sentence is too long.

RESPONSE TO REVIEWER 2, COMMENT #73

Sentence has now been broken up into 2 sentences and now reads thus:

" With respect to the degradation of these polymer matrices in aqueous media and under cathodic polarizations in the range expected on galvanic coupling with metals, the vital factors with respect to the polymers need to be highlighted. These vital factors are most likely to be; the polymer molecular weight or molecular weight distributions, the types of "linkages" and endgroups present, and the possibility of increasing the cross-link density in thermosetting resins by reactive species (O2,O22—, OH, HO2) produced from cathodic process which can lead to embrittlement."

REVIEWER 2, COMMENT #74

728: ‘indicating’ instead of ‘indicates’

RESPONSE TO REVIEWER 2, COMMENT #74

Correction has now been effected by substituting "indicates" with "indicating"

REVIEWER 2, COMMENT #75

760 – now referring to graphite composites- disjoints the paper as previously referred to as carbon- another point of American vs English language issues noticed in paper

RESPONSE TO REVIEWER 2, COMMENT #75

The word "graphite" has now been replaced with "carbon" to maintain uniformity.

REVIEWER 2, COMMENT #76

774 not sure this sentence makes sense

RESPONSE TO REVIEWER 2, COMMENT #76

Sentence has now been modified for clarity to read thus: "Taylor and Humffray [204] studied oxygen reduction on glassy carbon electrodes in solutions of high pH (pH> 10) and reported effects on the mechanism based on the type of prior treatment (cathodic or anodic) given to the glassy carbon electrodes. The major difference in mechanism of oxygen reduction was that oxygen reduction to OH- rather than to peroxide is boosted at all potentials in glassy carbon electrodes given a prior anodic treatment."

REVIEWER 2, COMMENT #77

788: Capitalise ‘Can’

RESPONSE TO REVIEWER 2, COMMENT #77

"can" has now been capitalized and now  appears thus: "Can"

REVIEWER 2, COMMENT #78

‘We’ is used at various point in the text as this is too informal- also as a review paper it should be compiling research already published thus no ‘we’

RESPONSE TO REVIEWER 2, COMMENT #78

I think that the use of "We" is inevitable in some instances as this is a critical review.  Being a critical review, it might be necessary to agree, disagree  or have and express differences in perspectives expressed in literature based on information available to authors at the moment.

REVIEWER 2, COMMENT #79

805: Sentence beginning with ‘Absence..’ doesn’t make sense.

RESPONSE TO REVIEWER 2, COMMENT #79

Sentence has been re-phrased for clarity to read thus: "Absence of fibre surface roughening,  the presence of small gaps between fibre and matrix (both observed from scanning electron microscopy), and decreasing slope of the impedance magnitude plot coupled with decreasing phase angle peak are indicative of the "squaring effect" that is characteristic of porous electrodes [359-361]. "

REVIEWER 2, COMMENT #80

839: ‘ant’?

RESPONSE TO REVIEWER 2, COMMENT #80

"ant" has been corrected to "any"

REVIEWER 2, COMMENT #81

830+: A review should be a collection of published (peer-reviewed) work.

RESPONSE TO REVIEWER 2, COMMENT #81

We agree that a review should be a collection of published (peer-reviewed) work.  In agreement with this position Figures 7 and 8 and the discussions thereof have been removed from this manuscript as they are extracts from an up-coming publication. However, Figure 6 which is strictly speaking the only "yet to be peer-reviewed" material is retained as it is necessary to highlight the use of delta phase angles to monitor CFRP degradation. It  has been emphasized that this presentation is based on data treatment procedure that had undergone peer-review.

REVIEWER 2, COMMENT #82

Caption for Figure 9: (‘In black’ as opposed to ‘in back’).

RESPONSE TO REVIEWER 2, COMMENT #82

Correction has now been effected.

REVIEWER 2, COMMENT #83

956: Replace ‘Figure 1’ with ‘Figure 2’.

RESPONSE TO REVIEWER 2, COMMENT #83

Correction has now been effected. "Figure 1" has now been replaced with "Figure 2".

REVIEWER 2, COMMENT #84

956 specie should be species

RESPONSE TO REVIEWER 2, COMMENT #84

Correction has now been effected. " specie " has now been replaced with " species "

Round  2

Reviewer 1 Report

The revised version takes into account all comments raised by me in my previous review. Therefore the paper is now fit for publication, in my opinion. Nevertheless moderate English language polishing should still be done.